# Investigation of carbapenemase-encoding genes in *Burkholderia cepacia* and *Aeromonas sobria* isolates from nosocomial infections in Iraqi patients

Mushtak T. S. Al-Ouqaili[1]*, Rawaa A. Hussein[2], Bushra A. Kanaan[3], Ahmed T. S. Al-Neda[4]

1 Department of Microbiology, College of Medicine, University of Anbar, Anbar Governorate, Ramadi, Iraq, 2 Department of clinical laboratory sciences, College of Pharmacy, University of Anbar, Anbar Governorate, Ramadi, Iraq, 3 Department of Medical Microbiology, College of Medicine, University of Anbar, Anbar governorate, Ramadi City, Iraq, 4 Department of Field Crops, College of Agriculture, University of Anbar, Anbar Governorate, Ramadi City, Iraq

* Ph.dr.mushtak_72@uoanbar.edu.iq

## Abstract

*Burkholderia cepacia* and *Aeromonas sobria* are difficult to eradicate due to their innate resistance to a variety of medications, and cause various diseases. The aim of this study was to investigate the occurrence of carbapenemase genes and patterns of antibiotic resistance in isolates of *B. cepacia* and *A. sobria*. Randomly, 120 clinical specimens have been collected in patients with nosocomial infections. Selective media were used to culture ear swabs, urine, burns, wounds and cerebrospinal fluids. According to biochemical tests and the VITEK-2 system, 75 of these demonstrated positive growth with *B. cepacia* and *A. sobria*. Metallo-β-lactamase (MBL) synthesis was phenotypically screened using the meropenem-EDTA disc test. The *recA* gene in *B. cepacia* and the genes encoding carbapenemase in both species were found using PCR tests. Among the 75 isolates assessed 20 (26.6%) were *A. sobria* and 55 (73.3%) were *B. cepacia*. Piperacillin, cefepime, and ceftriaxone showed antimicrobial resistance of 100%, followed by ceftazidime (97.3%), cefazolin (96%), and piperacillin/ tazobactam (94.6%). Intermediate resistance was reported with aztreonam (61.3%), meropenem (49.3%), trimethoprim-sulfamethoxazole (49.3%), gentamicin (46.6%), levofloxacin (44%), and ciprofloxacin (44%). It is important to note minocycline (40%), amikacin (40%) imipenem (36%) and tigecycline (34.6%), had the lowest resistance rates, hence their relatively higher efficacy against the tested isolates. In this investigation, the *B. cepacia* was confirmed to be found via the *recA* gene. The overall prevalence of carbapenemase genes was 92.8% (52/56) with $bla_{KPC}$ accounting for 80.8% (42/52) and $bla_{GES}$ for 19.2% (10/52) of the total. Specifically, 38 (90.51%) of the 42 (76.36%) *B. cepacia* isolates that were positive in carbapenem resistance carried $bla_{KPC}$ gene, 2 (4.81%) isolates carried $bla_{GES}$, and

**Data availability statement:** All relevant data are within the manuscript.

**Funding:** The author(s) received no specific funding for this work.

**Competing interests:** The authors have declared that no competing interests exist.

2 (4.81%) had no detectable carbapenemase gene. In the case of the 14 *A. sobria* carbapenem-resistant isolates, there were 4 isolates (28.6%) that had $bla_{KPC}$, 8 isolates (57.1%) that had $bla_{GES}$ and there were 2 isolates (14.3%) that did not have any carbapenemase genes. None of isolates studied tested positive for the $bla_{IMP}$ gene. The recent study concluded that *recA* gene identification was more sensitive and specific technique for detection *B. cepacia complex* isolates. Since the prevalence of carbapenemase producers is high, careful infection control measures, rapid diagnostics, and antimicrobial stewardship must be implemented by clinicians. It is necessary that combination therapy be guided and early detectable to ensure better outcomes and restrict resistance.

## Introduction

*Burkholderia cepacia* is a Gram-negative, rod-shaped, motile bacterium belonging to the Burkholderiaceae family (formerly known as *Pseudomonas cepacia*).Usually ranging in size from 1.6 to 3.2 µm, these organisms have attachment pili and multitrichous polar flagella. Commonly occurring in soil, water, and plants, *B. cepacia* can colonize immunocompromised individuals and result in serious opportunistic infections [1]. It remains to be a major and continuous impact in both medical facilities and community settings because of its inherent resistance to numerous antibiotics, capacity to survive in hostile circumstances, and ease of transmission [2]. According to Wigfield et al. (2002), the primary cellular resistance mechanisms that are primarily involved in the overall process of antibiotic resistance development in *B. cepacia* species include altered medication targets, enzymatically changed antimicrobials, permeability of the outer membrane (OMP), and efflux pumps [3]. *B. cepacia* isolates frequently show intrinsic resistance to several antibiotics or drugs classes, making infections difficult to treat. This resistance, which includes decreased susceptibility to substances like ticarcillin, cephalosporins, phosphonic acid antibiotics, polymyxins, and aminoglycosides, develops spontaneously and does not require gene acquisition or mutation [4]. Furthermore, Nzula et al. (2002) found that the *B. cepacia* complex cells vary in their inherent antimicrobial resistance models, which is most likely related to the genomovar type [5].

Aeromonas species are gram-negative heterotrophic bacteria that are primarily found in warm areas and can contaminate water, seafood, meat, and vegetables, and which can lead to human illness [6]. Aeromonas infections range in frequency from 20 to 76 cases per 1,000,000 people. *A. hydrophila, A. sobria,* and *A. caviae* are the most frequently isolated species [7]. These bacteria can cause a wide range of infections in humans, including of the gastrointestinal system, septicemia, acute respiratory tract infections, and soft tissue infections. Despite the fact that immunocompromised people are more likely to contract the disease, immunocompetent hosts can also contract it [8]. The current study focuses on the prevalence of *A. sobria*, highlighting its increasing clinical significance and justifying its inclusion in this analysis, despite the fact that *A. hydrophila* is more commonly reported in clinical infections.

Because Aeromonas infections have at least three chromosomal β-lactamases, a major worry with such is their possible resistance to penicillin, ampicillin, carbenicillin, and cefazolin. Aeromonas spp. have been shown to produce Ambler class B, C, and D β-lactamases. The main β-lactamases that Aeromonas harbors are penicillinase, metallo-β-lactamases (MBL), and AmpC β-lactamases [9,10]. Worldwide, nosocomial infections are a problem for both developed and low-resource nations [11]. One of the main reasons for increased mortality and morbidity amongst people who have been hospitalized is infections contracted in healthcare facilities; they pose a serious risk to public health in general, as well as the sufferer in particular [12]. Although *A. sobria* and *B. cepacia* are becoming more widely recognized as important nosocomial infection pathogens, little is known about the molecular resistance mechanisms of these bacteria, especially in relation to carbapenemase genes. More research on the resistance characteristics of multidrug-resistant bacteria is essential due to their rising prevalence. The aim of this study was to investigate the occurrence of carbapenemase genes and patterns of antibiotic resistance in isolates of *B. cepacia* and *A. sobria*.

## Materials and methods

### Ethics statement

The administrative human Ethical Approval Committee of the University of Anbar in Ramadi, Iraq, approved the study methods concerning the patients correspondingly to the official order numbered 91 in 05-1-2024. During the study, the willingness of all participants to participate it was determined according to the Helsinki Declaration of 2013. Informed written consent was provided by all patients (or their parents) participating in the study.

### Collecting and processing specimens

A total of 120 patients who attended the Ramadi Teaching Hospitals in the Al-Anbar Governorate between January and July 2024 were chosen at random to provide specimens. The participants were 50 women (41.7%) and 70 men (58.3%), with ages ranging from 7 to 70. Participants in the study were specifically chosen based on their age, clinical status, and the kind of specimens that were collected, in addition to additional inclusion criteria. All patients had an equal chance of being included, and bias was reduced by using the random sampling procedure. Patients with suspected nosocomial infections provided a total of 120 clinical specimens, which were chosen at random. The inclusion criteria focused on specimens from patients exhibiting clinical indications of infection, especially those with immunosuppression or cystic fibrosis, which are known risk factors for infections with *B. cepacia* and *A. sobria*. Clinically relevant samples such as cerebrospinal fluid (CSF), urine, ear swabs, diabetic foot ulcers, blood, and wound swabs were used to collect specimens. Samples that showed signs of environmental contamination or colonization were not included. MacConkey agar, blood agar, and mannitol salt agar (Oxoid, UK) were used to cultivate each specimen. They were then incubated aerobically at 37°C for 18–24 hours while maintaining sterility.

### *B. cepacia* and *A. sobria* isolate identification

Gram-negative isolates suspected of being *A. sobria* or *B. cepacia* were initially identified using normal bacteriological techniques. According to MacFaddin's (2000) procedures, these included Gram staining, oxidase and motility tests, and growth assessment on selective medium such MacConkey agar, blood agar, and mannitol salt agar [13]. Morphological and microscopic features were also assessed. The VITEK® 2 Compact B System (BioMérieux, France) with VITEK® 2 GN ID cards was used to confirm the final species-level identification in accordance with the manufacturer's instructions. Subsequent to this procedure, pure isolates were stocked in 20% glycerol incorporated in BHI (Oxoid, UK).

### Antibiotic susceptibility

According to the criteria defined by the Clinical and Laboratory Standards Institute (CLSI, 2024) the antibiotic susceptibilities of *B. cepacia* and *A. sobria* isolates were tested using the VITEK-2 System (BioMérieux, Marcyl'Étoile, France)

 

[14]. The antibiotics used included piperacillin, piperacillin/tazobactam, cefazolin, ceftazidime, ceftriaxone, cefepime, imipenem, meropenem, aztreonam, minocycline, amikacin, levofloxacin, ciprofloxacin, tigecycline, gentamicin, and trimethoprim-sulfamethoxazole.

## Methods for phenotypic identification of Metallo-β-Lactamase (MBL) production

A total of 56 strains resistant to at least one carbapenem were investigated. In this experiment, a disc containing meropenem (10 µg) and meropenem-EDTA (MBL inhibitor solution (0.5 M EDTA) (10/750 µg)) was applied to the lawn culture of the test strain's 0.5 McFarland inoculum. After incubating for 16–18 hours at 35°C, the zones of inhibition of the two discs showed a difference in diameter of ≥ 7 mm, which suggested MBL formation [15].

## DNA extraction

*B. cepacia* and *A. sobria* isolates were grown overnight in brain heart infusion broth under completely aseptic conditions to obtain the purified single colonies. Genomic DNA was extracted after incubation with DNA extraction kits of commercial origin (Sacace, Italy), in accordance with the manufacturer-provided guidelines [15,16]. All nucleic acids were collected and frozen at below -20°C using a deep freezer. The PCR technique was then utilized to search for, and identify, all the genes listed in Table 1.

## PCR techniques for *recA* gene detection in *B. cepacia* isolates

Using the *recA* gene of the Bcc complex in clinical samples and BCR1 and BCR2 primers, a standard PCR technique was used to directly molecularly identify the Bcc complex in all 75 extracted DNA samples (Table 1). A 25 µl reaction system comprising 12.5 µl of GoTaq® Green Master Mix (2X) (Promega, USA), 1 µl of each forward and reverse primer, 7.5 µl of nuclease-free water, and 3 µl of extracted DNA template was used for the PCR experiments. For the *recA* gene, the PCR reaction protocol was set up as follows: 30 cycles of 45 seconds at 94 ° C, 45 seconds at 56 ° C, and 90 seconds at 72 ° C were followed by one cycle of 94 °C for two minutes. At 72°C, the last extension step was set for a single 7-minute cycle. Using a molecular size measurement of one hundred base pairs (from Bioneer, Korea), the PCR products were seen on a 1.5% agarose gel stained with red safe nucleic acid staining (Intron, Korea) [17].

## Molecular methods for identification of carbapenemase genes

12.5 µl of GoTaq® Green Master Mix (2X), 1 µl of forward and reverse primers (Macrogen, Korea), 3 µl of bacterial DNA, 0.5 µl of $MgCl_2$, and 7 µl of double distilled and deionized water made up a total volume of 25 µl. The following conditions for PCR (Polymerase Chain Reaction) cycling were employed: 30 cycles of one minute each at 95°C, one minute at 50 or 55°C, one minute at 72°C, and ten minutes at 72°C (as indicated by Table 2). The PCR (Polymerase Chain Reaction)

Table 1. The specific primers are used in this investigation.

| Gene Name | Primer | Sequence (5′–3′) | Product Size (bp) | Annealing Temp (°C) | Reference |
|---|---|---|---|---|---|
| *B. cepacia complex recA* | BCR1 | TGACCGCCGAGAAGAGCAA | 1043 | 56 | [17] |
| | BCR2 | CTCTTCTTCGTCCATCGCCTC | | | |
| *bla*$_{KPC}$ | Forward (F) | ATG TCA CTG TAT CGC CGT CT | 345 | 55 | [4] |
| | Reverse (R) | TTT TCA GAG CCT TAC TGC CC | | | |
| *bla*$_{IMP}$ | Forward (F) | TTGACACTCCATTTACDG | 139 | 50 | [4] |
| | Reverse (R) | GATYGAGAATTAAGCCACYCT | | | |
| *bla*$_{GES}$ | Forward (F) | ATGCGCTTCATTCACGCAC | 864 | 55 | [18] |
| | Reverse (R) | TAATCAGTGAGGCACCTATCTC | | | |

Table 2. The distribution of study bacteria in relation to sample type, age, and gender of the subjects.

| Demographic Characteristics | Bacterial Isolates | | | | Total (n, %) | Mean Age±SD years |
|---|---|---|---|---|---|---|
| | B. cepacia | | A. sobria | | | |
| | n. | % | n. | % | | |
| Gender | | | | | | |
| Male | 38 | 50.7% | 6 | 8% | 44 (58.7%) | 37.5±16.16 |
| Female | 17 | 22.6% | 14 | 18.7% | 31 (41.3%) | 44.0±15.30 |
| Age (years) | | | | | | |
| ≤10 | 5 | 6.7% | 1 | 1.3% | 6 (8%) | 9.00±1.26 |
| 11–20 | 1 | 5.3% | 2 | 2.6% | 3 (4%) | 16.33±4.73 |
| 21–30 | 14 | 18.6% | 8 | 10.7% | 22 (29.3%) | 25.36±2.93 |
| 31–40 | 9 | 12% | 1 | 1.3% | 10 (13.3%) | 35.5±3.03 |
| 41–50 | 1 | 1.3% | 3 | 4% | 4 (5.3%) | 44.25±4.03 |
| 51–60 | 3 | 4% | 1 | 1.3% | 4 (5.3%) | 55.5±4.2 |
| > 60 | 22 | 29.3% | 4 | 5.3% | 26 (34.6%) | 65.0±2.78 |
| Type of sample | | | | | | |
| Wound swab | 26 | 34.6% | 8 | 10.6% | 34 (45.3%) | 31.5±7.8 |
| Urine | 18 | 24% | 7 | 9.3% | 25 (33.3%) | 47.5±13.0 |
| Ear swab | 1 | 1.3% | 1 | 1.3% | 2 (2.6%) | 16.5±9.19 years |
| Cerebrospinal fluid | 2 | 2.6% | 0 | 0% | 2 (2.6%) | 12.5±2.5 |
| Diabetic foot ulcer | 5 | 6.7% | 2 | 2.7% | 7 (9.3%) | 13.14±2.73 |
| Blood | 3 | 4% | 2 | 2.7% | 5 (6.7%) | 16.4±8.76 |
| Total | 55 | (73.3%) | 20 | (26.6%) | 75 (100%) | 40.13±22.5 |

P-value = 0.422 non-significant.

product was analyzed using a 100 bp molecular size marker gel and detected using Red Safe Nucleic Acid Staining [16] (Intron, Korea). Table 1.

## Statistical analysis

The SPSS™ program, version 22.0 (IBM Corporation, Armonk, NY, USA), was used to analyze the data. The chi-squared test was employed for analysis, and descriptive statistics such as mean and percentage were employed. The Wilson score method was used to calculate the 95% CI of the proportion. A p-value of under 0.05 was considered significant.

## Results

### Features and demographics of the isolated bacteria

Patients with suspected nosocomial infections provided a total of 120 clinical specimens, which were chosen at random. 75 (62.5%) of the 120 specimens showed positive growth for A. sobria and B. cepacia. Among the 75 culture positive isolates evaluated in this study 20 (26.6%) were identified to be A. sobria and 55 (73.3%) were B. cepacia. Wound swabs accounted for the majority of isolates (45.3%), followed by urine samples (33.3%), diabetic foot ulcers (9.3%), and other clinical sources (Table 3). The six (6) different sample types were used to obtain the study microorganisms. In terms of the gender distribution of the individuals from whom the isolates were collected, 44 (58.7%) were male and 31 (41.3%) female (see Table 2). Average age of the study participants (between the ages of 7–70 years) who had tested positive to the study isolates was 40.13±22.5 years. Patients over 60 years old accounted for 34.6% of the isolates, followed by those between the ages of 21 and 30 (29.3%). The patients from whom the fewest specimens taken (4%) were those between the ages of 11 and 20.

**Table 3. Findings regarding the phenotypic and genotypic methods for identifying carbapenem resistance.**

| Isolates | Phenotype | | | Genotype | | |
|---|---|---|---|---|---|---|
| | Disc | Double disk synergy test | PCR | *blaKPC* | *blaIMP* | *blaGES* |
| | n = 75 | n = 56 | n = 56 | | | |
| *B. cepacia* | n = 42 (56%) | n = 39(69.6%) | 40 (71.4%) | 38 (67.9%) | – | 2 (3.6%) |
| *A. sobria* | n = 14 (18.6%) | n = 10 (17.9%) | 12 (21.4) | 4 (7.1%) | – | 8 (14.3%) |
| Total | n = 56 (74.6%) | n = 49 (87.5%) | 52 (92.8%) | 42 (75%) | – | 10 (17.9%) |

## Antimicrobial susceptibility

Piperacillin, cefepime, and ceftriaxone showed antimicrobial resistance of 100%, followed by ceftazidime (97.3%), cefazolin (96%), and piperacillin/ tazobactam (94.6%). Intermediate resistance was reported with aztreonam (61.3%), meropenem (49.3%), trimethoprim-sulfamethoxazole (49.3%), gentamicin (46.6%), levofloxacin (44%), and ciprofloxacin (44%). It is important to note minocycline (40%), amikacin (40%) imipenem (36%) and tigecycline (34.6%), had the lowest resistance rates, hence their relatively higher efficacy against the tested isolates. According to the VITEK-2 System, a total of 56 (74.6%) of the *B. cepacia* and *A. sobria* isolates were shown to exhibit carbapenem resistance. Of the 56 isolates, *B. cepacia* accounted for 75% (n = 42) and *A. sobria* for 25% (n = 14). Of the 56 isolates that tested positive for carbapenem resistance, 49.3% were resistant to meropenem and 36% to imipenem. Of the 42 isolates of *B. cepacia*, 47.6% (20/42; 95% CI: 33.0%–62.6%) showed resistance to imipenem, and 66.7% (28/42; 95% CI: 51.6%–79.1%) showed resistance to meropenem. However, among the 14 isolates of *A. sobria*, 50.0% (7/14; 95% CI: 26.0%–74.0%) were resistant to imipenem, while 64.3% (9/14; 95% CI: 38.8%–83.7%) were resistant to meropenem. There was no significant correlation between species and meropenem resistance, according to a chi-square test ($\chi^2 = 0.169$, p = 0.681), suggesting that the two species' resistance rates were comparable. Similarly, imipenem resistance and species did not show a statistically significant correlation ($\chi^2 = 0.034$, p = 0.853), indicating similar resistance levels. Furthermore, of the 56 isolates resistant to carbapenem, 38 (67.8%) were collected from male patients, and 28 (50%) from female. The Chi-square test statistical analysis showed a significant association between isolates' carbapenem resistance and patient gender ($\chi^2 = 7.52$, p = 0.006). In isolates from male patients, resistance was significantly higher (86.4%, 95% CI: 74.5–94.1%) than in those from female patients (58.1%, 95% CI: 39.1–75.5%). The range of carbapenem MICs in this investigation was 0.2 µg/ml to 64 µg/ml. The results for meropenem MIC (which indicated that 36 isolates were resistant to carbapenem whilst 18 isolates were susceptible) appeared to be a more accurate indicator of the presence of the carbapenemase gene than the results for imipenem (which showed that 29 isolates with the carbapenemase gene were sensitive to carbapenem). Additionally, all seven isolates that harbored the carbapenemase gene were resistant to carbapenem. Profiles of drugs resistance are illustrated in Fig 1.

Depending on MICs, the antimicrobial susceptibility surveillance was detected for piperacillin (PIP), piperacillin/tazobactam (TZB), ceftazidime (CAZ), cefazolin (CFZ) cefepime (FEP), ceftriaxone (CRO), aztreonam (ATM), meropenem (MEM), imipenem (IMP), minocycline (Min) amikacin (AK), levofloxacin (LVX), ciprofloxacin (CIP), tigecycline (TGC), gentamicin (GI10), and trimethoprim-sulfamethoxazole (SXT).

## *recA* gene detection in *B. cepacia* isolates

The *B. cepacia* was identified in this study using the *recA* gene. The *recA* gene was positive in all 55 isolates that the VITEK 2 system classified as *B. cepacia*, confirming their inclusion in the *B. cepacia* complex, as shown in Fig 2. Four isolates were found to be *B. cepacia* complex by VITEK 2 System. Two isolates tested negative for *recA* gene and were found to be *B. cepacia* by VITEK 2 System.

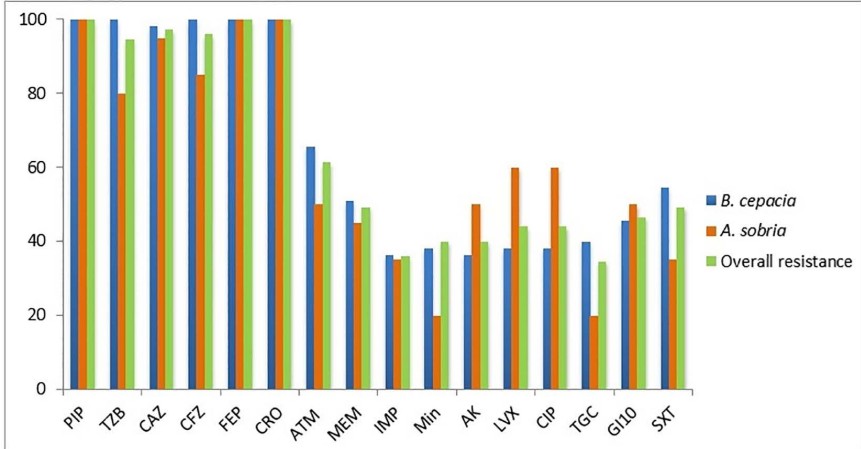

**Fig 1. Patterns of drug resistance among the bacteria considered in the investigation.**

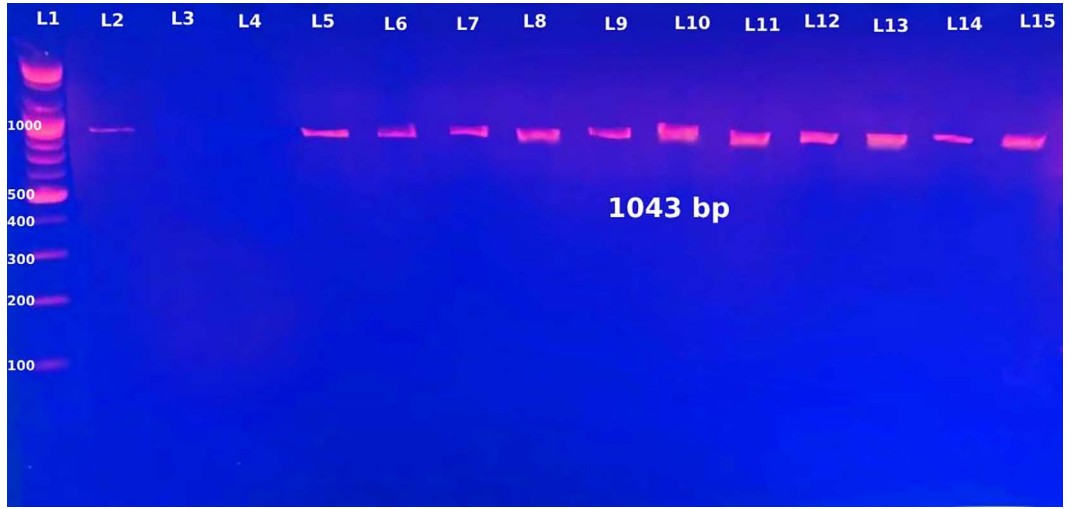

**Fig 2. The *recA* gene's gel electrophoresis (1043 base pairs). The lanes (2,4,5,6,7,8,9) were used to load the DNA samples. L3 and L4 showed negative results for *recA* gene. In lane 1, the molecular weight marker was the 100 base pair DNA ladder.**

### Phenotypic and Genotypic Validation of the Synthesis of Carbapenemase and the Occurrence of Gene Encoding for Carbapenemase in Isolates of *B. cepacia* and *A. sobria*

In terms of the phenotypic validation of carbapenem resistance, the double disk synergy test yielded positive results for 87.5% (49/56) of the isolates that were resistant to meropenem and imipenem via double disk synergy (Table 3 and Fig 3). The molecular detection analysis was performed by testing 55 (73.33%) *B. cepacia* and 20 (26.66%) *A. sobria* isolates DNA in search of the possession of carbapenemase genes by PCR (Polymerase Chain Reaction). The double disk synergy test and PCR were used to determine the meropenem and imipenem resistance characteristics as mediated by the $bla_{KPC}$ and $bla_{GES}$ genes. No evidence for the $bla_{IMP}$ gene was found. Tables 4 and 5 report the distribution of the various carbapenemase genes. PCR was used in this investigation to identify the $bla_{KPC}$-, $bla_{IMP}$-, and $bla_{GES}$-like genes among the 56 isolates resistant to carbapenem. Using PCR, the overall prevalence of carbapenemase genes was

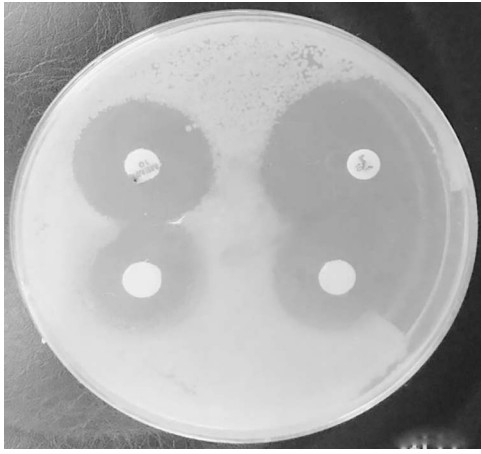

**Fig 3. Phenotypic preliminary test for detection of carbapenemase production.**

found to be 92.8%% (52/56) with $bla_{KPC}$, accounting for 80.8% (42/52), and $bla_{GES}$, for 19.2% (10/52) of the total. Specifically, 38 (90.51%) of the 42 (76.36%) *B. cepacia* isolates that were positive in carbapenem resistance carried $bla_{KPC}$ gene, 2 (4.81%) isolates carried $bla_{GES}$, and 2 (4.81%) had no detectable carbapenemase gene. In the case of the 14 *A. sobria* carbapenem-resistant isolates, there were 4 isolates (28.6%) that had $bla_{KPC}$, 8 isolates (57.1%) that had $bla_{GES}$ and there were 2 isolates (14.3%) that did not have any carbapenemase genes as can be seen in Figs 4, 5. None of the isolates had more than one gene of carbapenemase at the same time. None of the isolates contained the $bla_{IMP}$ gene. The presence of carbapenemase genes ($bla_{KPC}$, $bla_{GES}$), as well as the lack of any carbapenemase gene, was significantly correlated with carbapenem resistance in isolates of *B. cepacia* and *A. sobria*, according to the Chi-square test for independence ($x^2 = 22.83$, p = 0.05). In particular, isolates with the $bla_{KPC}$ gene (90.5%, 95% CI: 85.1–95.0%) exhibited considerably higher resistance than isolates with the $bla_{GES}$ gene (66.7%, 95% CI: 41.3–88.3%) or without the carbapenemase gene (50.0%, 95% CI: 31.5–68.5%). These results underline the crucial role that carbapenemase genes play in resistance mechanisms by indicating a substantial correlation between their existence and carbapenem resistance. Both phenotypic and genotypic techniques were used in this investigation to evaluate the synthesis of carbapenemase by isolates of *B. cepacia* and *A. sobria*. Although there was a high level of consistency, a subset of isolates showed differences between the two methods. To be specific, some isolates showed phenotypic carbapenemase activity using the meropenem-EDTA combined disc synergy test, although PCR results for the targeted genes ($bla_{KPC}$, $bla_{IMP}$, and $bla_{GES}$) were negative. On the other hand, other isolates have carbapenemase genes but showed little to no activity in the phenotypic test. 24 wound samples, 14 urine samples, and two CSF fluids tested positive for the total of 40 $bla_{KPC}$-positive *B. cepacia*. $bla_{GES}$, however, were found in the isolates gained from two diabetic foot ulcers. Four $bla_{KPC}$-positive *A. sobria* were isolated from two samples from wounds and one sample each from urine and ears. $bla_{GES}$ were identified in urine (two), blood (three), and diabetic foot ulcers (three). The statistical analysis showed that resistance to imipenem, meropenem, cefepime, ceftriaxone, piperacillin (P-value = 0.000), piperacillin/tazobactam (P-value = 0.003), ceftazidime (P-value = 0.001), and cefazolin (P-value = 0.002) was significantly correlated with the presence of carbapenemase genes. The range of carbapenem MICs in this investigation was 0.2 µg/ml to 64 µg/ml.

## Discussion

The present study offers significant new information on the distribution, carbapenemase gene profiles, and antimicrobial resistance (AMR) patterns of isolates of *B. cepacia* and *A. sobria* from clinical specimens in Anbar Governorate,

**Table 4. Antibiogram and resistance genes of isolates of *B. cepacia* that were determined to be carbapenemase positive.**

| Isolate No | Isolates | PCR | Carbapenems (MIC–µg/ml) | |
|---|---|---|---|---|
| | | | **MEM** | **IMP** |
| 1 | *Burkholderia cepacia* | $bla_{KPC}$ | R (16) | R (16) |
| 3 | *Burkholderia cepacia* | $bla_{KPC}$ | R (32) | R (64) |
| 7 | *Burkholderia cepacia* | $bla_{KPC}$ | R (32) | R (32) |
| 8 | *Burkholderia cepacia* | $bla_{KPC}$ | R (16) | R (32) |
| 9 | *Burkholderia cepacia* | $bla_{KPC}$ | R (16) | R (16) |
| 10 | *Burkholderia cepacia* | $bla_{KPC}$ | R (64) | R (16) |
| 11 | *Burkholderia cepacia* | $bla_{KPC}$ | R (32) | S (0.5). |
| 12 | *Burkholderia cepacia* | $bla_{KPC}$ | R (32) | S (0.5). |
| 13 | *Burkholderia cepacia* | $bla_{KPC}$ | R (32) | S (0.5). |
| 14 | *Burkholderia cepacia* | $bla_{KPC}$ | R (16) | S (0.5). |
| 20 | *Burkholderia cepacia complex* | $bla_{KPC}$ | R (16) | S (4) |
| 21 | *Burkholderia cepacia complex* | $bla_{KPC}$ | R (32) | S (4) |
| 22 | *Burkholderia cepacia complex* | $bla_{KPC}$ | R (32) | S (0.2). |
| 23 | *Burkholderia cepacia complex* | **blaGES** | R (16) | S (0.2). |
| 24 | *Burkholderia cepacia* | $bla_{KPC}$ | R (16) | S (0.2). |
| 25 | *Burkholderia cepacia* | $bla_{KPC}$ | R (64) | S (4) |
| 26 | *Burkholderia cepacia* | $bla_{KPC}$ | R (16) | S (0.2). |
| 27 | *Burkholderia cepacia* | $bla_{KPC}$ | R (16) | S (4) |
| 28 | *Burkholderia cepacia* | $bla_{KPC}$ | R (16) | S (4) |
| 29 | *Burkholderia cepacia* | $bla_{KPC}$ | R (32) | S (4) |
| 35 | *Burkholderia cepacia* | $bla_{KPC}$ | R (16) | S (0.2). |
| 36 | *Burkholderia cepacia* | $bla_{KPC}$ | R (32) | S (0.2). |
| 37 | *Burkholderia cepacia* | **blaGES** | R (64) | S (0.2). |
| 38 | *Burkholderia cepacia* | $bla_{KPC}$ | R (16) | S (0.2). |
| 39 | *Burkholderia cepacia* | $bla_{KPC}$ | R (16) | S (0.2). |
| 40 | *Burkholderia cepacia* | $bla_{KPC}$ | R (16) | S (0.2). |
| 41 | *Burkholderia cepacia* | $bla_{KPC}$ | R (32) | S (0.5). |
| 42 | *Burkholderia cepacia* | $bla_{KPC}$ | R (32) | S (0.5). |
| 43 | *Burkholderia cepacia* | $bla_{KPC}$ | S (0.5). | R (16) |
| 44 | *Burkholderia cepacia* | $bla_{KPC}$ | S (0.2). | R (16) |
| 45 | *Burkholderia cepacia* | $bla_{KPC}$ | S (4) | R (16) |
| 46 | *Burkholderia cepacia* | $bla_{KPC}$ | S (1) | R (16) |
| 47 | *Burkholderia cepacia* | $bla_{KPC}$ | S (0.5). | R (16) |
| 48 | *Burkholderia cepacia* | $bla_{KPC}$ | S (0.5). | R (16) |
| 49 | *Burkholderia cepacia* | $bla_{KPC}$ | S (0.5). | R (16) |
| 50 | *Burkholderia cepacia* | $bla_{KPC}$ | S (0.5). | R (32) |
| 51 | *Burkholderia cepacia* | $bla_{KPC}$ | S (0.2). | R (32) |
| 52 | *Burkholderia cepacia* | $bla_{KPC}$ | S (0.2). | R (32) |
| 53 | *Burkholderia cepacia* | $bla_{KPC}$ | S (4) | R (32) |
| 54 | *Burkholderia cepacia* | $bla_{KPC}$ | S (1) | R (32) |
| 55 | *Burkholderia cepacia* | $bla_{KPC}$ | S (0.2). | R (64) |
| 56 | *Burkholderia cepacia* | $bla_{KPC}$ | S (0.2). | R (64) |

*MEM: Meropenem, IMP: Imipenem

*PCR: Polymerase Chain Reaction detection of resistance genes

*S: Susceptible, R: Resistant

*MIC: Minimum Inhibitory Concentration in µg/mL

$bla_{GES}$, $bla_{KPC}$: Detected carbapenemase genes

**Table 5. Antibiogram and resistance genes of carbapenemase producing isolates of *A. sobria*.**

| Isolate No | Isolates | PCR | Carbapenems (MIC–µg/ml) | |
|---|---|---|---|---|
| | | | **MEM** | **IMP** |
| 2 | *Aeromonas sobria* | $bla_{GES}$ | S (0.2). | R (32) |
| 4 | *Aeromonas sobria* | – | S (0.5). | R (32) |
| 5 | *Aeromonas sobria* | $bla_{GES}$ | S (0.5). | R (16) |
| 6 | *Aeromonas sobria* | $bla_{GES}$ | S (4) | R (16) |
| 15 | *Aeromonas sobria* | – | R (16) | R (16) |
| 16 | *Aeromonas sobria* | $bla_{KPC}$ | R (16) | R (32) |
| 17 | *Aeromonas sobria* | $bla_{GES}$ | S (0.5). | R (64) |
| 18 | *Aeromonas sobria* | $bla_{GES}$ | R (32) | S (0.5). |
| 19 | *Aeromonas sobria* | $bla_{KPC}$ | R (32) | S (0.5). |
| 30 | *Aeromonas sobria* | $bla_{GES}$ | R (32) | S (0.5). |
| 31 | *Aeromonas sobria* | $bla_{KPC}$ | R (32) | S (0.2). |
| 32 | *Aeromonas sobria* | $bla_{GES}$ | R (64) | S (0.2). |
| 33 | *Aeromonas sobria* | $bla_{GES}$ | R (16) | S (1) |
| 34 | *Aeromonas sobria* | $bla_{KPC}$ | R (16) | S (1) |

*MEM: Meropenem, IMP: Imipenem

*PCR: Polymerase Chain Reaction detection of resistance genes

*S: Susceptible, R: Resistant

*MIC: Minimum Inhibitory Concentration in µg/mL

$bla_{GES,}$ $bla_{KPC}$: Detected carbapenemase genes

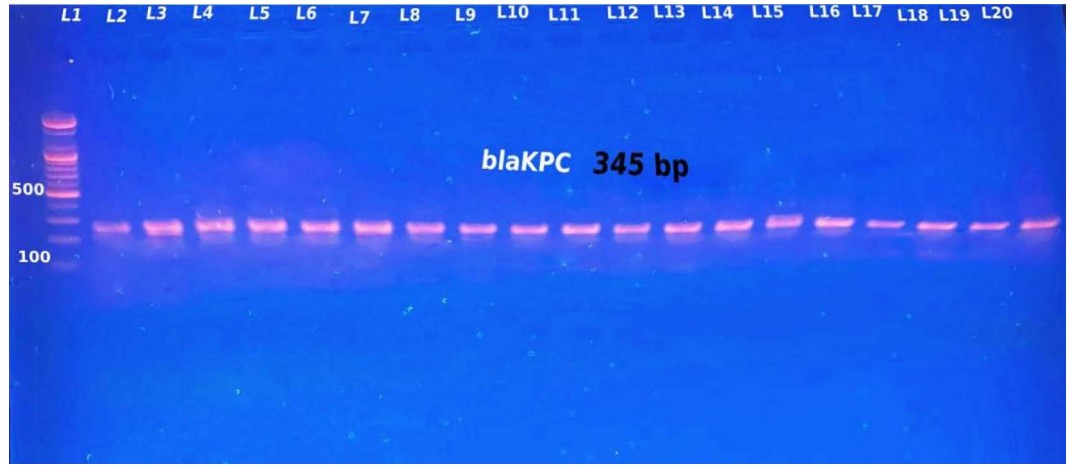

**Fig 4. Positive results for the *blaKPC* gene were obtained via 1.5% agarose gel electrophoresis.** The gene sizes in lanes 1–20 were found to be 345 bp. A DNA ladder in the first lane functioned as a 100 bp molecular weight marker.

Iraq. Particularly in individuals with compromised immune systems, *B. cepacia* is known to be a multidrug-resistant opportunistic pathogen, which is why it predominates over *A. sobria* (26.6%). Notably, male patients were more likely to have *B. cepacia* isolated from them, which is in line with gender distributions that have been previously documented [17,18]. *A. sobria* is also almost equally distributed across male and female patients, which confirms previous findings

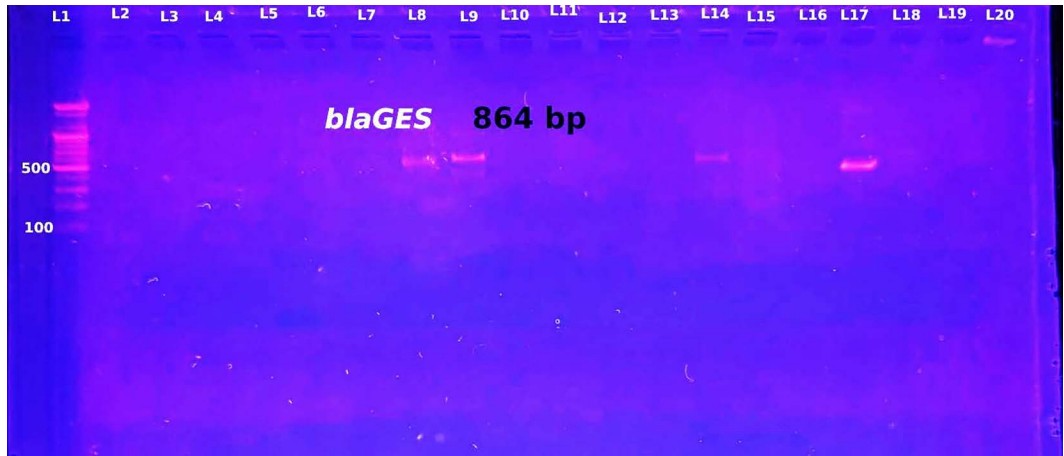

**Fig 5. Positive results for the *blaGES* gene were obtained via 1.5% agarose gel electrophoresis.** In lanes 8, 9, and 17, the gene sizes were 864 bp. A DNA ladder in lane 1 functioned as a 100 bp molecular weight marker.

by Pineda-Reyes et al. [19] and raises the possibility that host susceptibility differences depending on gender may not be important for this organism. The recognized tissue tropism of these pathogens for skin and urinary tract infections is reflected in the prevalence of wound swabs and urine specimens as sources for isolation. The fact that *B. cepacia* has been recovered from a variety of clinical samples, ranging from diabetic foot ulcers and cerebrospinal fluid, is in line with its well-established adaptability as an opportunistic pathogen [20]. The isolation of *A. sobria* from blood, wounds, and urine, on the other hand, supports its increasing medical significance, especially in hospital settings, even though it was not found in cerebrospinal fluid samples [21]. These results highlight both species' clinical significance, especially in light of their high levels of first-line antibiotic resistance. These resistance patterns have important ramifications, particularly in areas with few empirical treatment alternatives and little utilization of laboratory-guided therapy.

Options for therapy with *B. cepacia* remain limited. Application of restricted medications such as Clavulanic acid, ceftazidime, meropenem, minocycline, levofloxacin, chloramphenicol, and trimethoprim-sulfamethoxazole is advised by the CLSI guide [11]. While resistances to ceftazidime and trimethoprim/sulfamethoxazole were 98.18% (54/55) and 54.54 (30/55), respectively, susceptibility to meropenem and minocycline remained more prevalent, with 40% (22/55) and 38.18% (26/55) of isolates being resistant, respectively. Al-Muhanna et al. [4] and Tuwaij et al. [22] have documented considerable carbapenem resistance, and our results show comparable resistance trends to those reported in other Middle Eastern nations. The fact that they exceed the resistance levels observed in Ghana and other parts of the world [23], however, indicates that antimicrobial usage practices and resistance evolution vary by place. These patterns emphasize how crucial localized monitoring data is for informing treatment recommendations and how strong antimicrobial stewardship initiatives are essential. These included: 100% for ceftriaxone, cefoxitin, and cefepime; 87.5% for ticarcillin with clavulanic acid, piperacillin, ceftazidime, tobramycin, ciprofloxacin, and levofloxacin; 62.5% for aztreonam and amikacin; and 37.5% for meropenem, imipenem, and gentamicin. The isolates of *A. sobri* in this investigation demonstrated little resistances to minocycline and tigecycline; 20% and 35%, respectively, to meropenem and imipenem; and high resistances to piperacillin, cefepime, and ceftriaxone (100%), and to ceftazidime (95%). The present study was in line with Rhee et al., who observed high antibiotic resistance (15.5%) for CRO and TZP, but only 9.8% and 3.0%, respectively, to amikacin and carbapenem [24]. In this work, phenotypic screening for carbapenemase production was conducted using the double-disk synergy test (DDST). Despite not being the gold standard per CLSI recommendations, DDST is nonetheless a useful and popular technique in environments with limited resources because of its low cost, simplicity of use, and capacity to detect the presence of certain serine carbapenemases and metallo-β-lactamases.

In this work, *B. cepacia* and *A. sobria* isolated from clinical specimens in Anbar Governorate, Iraq, were evaluated for the prevalence of carbapenem resistance, AMR patterns, and carbapenemase gene distribution. This study is amongst the few, rare carbapenem resistance studies to consider this country. Resistances to meropenem and imipenem were 40% and 36.36% in *B. cepacian*, which appear to be higher than was found in by Tuwaij et al. [22]. For *A. sobria*, the frequency of meropenem and imipenem resistance was 35%, which was also greater than observed [24]. The resistance profiles found in this investigation highlight important treatment challenges, especially for infections caused by *A. sobria* and *B. cepacia*. The high rates of resistance to routinely used β-lactam antibiotics, such as trimethoprim/sulfamethoxazole and ceftazidime, severely restrict the alternatives for empirical treatment. For *B. cepacia*, a bacterium frequently found in individuals with immunocompromised or cystic fibrosis, prompt and efficient antibiotic therapy is crucial to preventing negative effects. Although limited, the observed susceptibility to minocycline and meropenem may provide alternate therapeutic options; nonetheless, the emergence of resistance to these medicines (about 40 and 38.2%, respectively) emphasizes the critical need for susceptibility-guided therapy.

The first step in classifying an isolate within the *B. cepacia* complex is to amplify *recA* using primers BCR1 and BCR2. The *B. cepacia* was identified in this study using the *recA* gene. The *recA* gene was positive in all 55 isolates that the VITEK 2 system classified as *B. cepacia*, confirming their inclusion in the *B. cepacia* complex. Four isolates were found to be *B. cepacia* complex by VITEK 2 System. Two isolates tested negative for *recA* gene and were found to be B. cepacia by VITEK 2 System. Brisse and colleagues discovered that, despite the fact that *B. gladioli* is not listed in the VITEK2 database (and so should have a higher probability of misidentification), only one isolate of the bacteria was recognized by the VITEK 2 apparatus as "*B. cepacia* or *B. pseudomalleii*." However, as other isolates of these species or other non-fermenters like Achromobacter or Alcaligenes may be mistakenly recognized as *B. cepacia*, it is always an excellent choice to use a molecular approach to validate the *B. cepacia* identification [25]. The *recA* gene has been widely used in bacterial systematics and has proven to be highly helpful in identifying species of the *B. cepacia* complex. By using phylogenetic analysis of sequence variance within the gene, it is possible to distinguish between the nine present species of the *B. cepacia* complex. BCR1 and BCR2, the original *recA*-based PCR primers, are exclusive to the *B. cepacia* complex members and do not amplify this gene in other *B. cepacia* species. This limits the technique's ability to classify other Burkholderia species in various natural habitats, even though it can be a useful method of confirming an isolate's position within the complex. The gene technique PCR (Polymerase Chain Reaction) results are highly accurate, and we can rely on them [26].

Carbapenemases are enzymes that have various different hydrolytic profiles; all are categorized as β-lactamases. All the β-lactam antibiotics, including the penicillins, cephalosporins, monobactams, and the carbapenems, are acted upon by these enzymes. Due to carbapenemase activity, a majority of β-lactam drugs can become inactive against the severe infections that result from bacteria that produce such β-lactamases. Enzyme superfamilies' comprise carbapenemases from the A, B, and D classes of β-lactamases. Zinc is present in the active region of class B enzymes, which are metallo-β-lactamases; class A and D enzymes, on the other hand, hydrolyze serine [23]. The Guiana-Extended-Spectrum and *Klebsiella pneumoniae* carbapenemase families are amongst the Class A carbapenemases. The KPC carbapenemases, which are mainly found on *Klebsiella pneumoniae* plasmids, are the most well-known members of this category. The OXA-type β-lactamases are classified as class D carbapenemases, and are mostly generated by *Acinetobacter baumannii* [27]. Five further subgroups include IMP, VIM, SPM, GIM, and SIM, and mainly belong to *Pseudomonas aeruginosa*. The characteristics, epidemiology, and detection of carbapenemases found in pathogenic bacteria have been updated in the present article [28,29]. Furthermore, the phenotypic confirmatory tests conducted in this study revealed that 87.5% (49/56) of the isolates had carbapenem resistance, namely via the double synergy test. Contrary to the current study, which concentrated on *B. cepacia* and *A. sobria* Codjoe et al. in Ghana revealed a significantly lower incidence of 18.9% via the phenotypic method, which may be because the majority of their isolates were *Acinetobacter* spp. and *Pseudomonas* spp [23]. Panduragan and associates, 2015 study in India indicated that 62% of isolates developed

carbapenemases when both the phenotypic and genotypic approaches were used in gram-negative bacteria [30]. This finding is consistent with the current study, which identified similar proportions (87.5% and 92.8%) using each of these methods. The PCR method used in this study did not detect all carbapenemase genes; hence, those that tested positive for $bla_{KPC}$ and $bla_{GES}$ but negative for other carbapenemase enzymes may process other types of carbapenemase enzyme. Certain bacterial isolates were found to produce carbapenemase enzymes both via the phenotypic method and PCR; other isolates were found to be negative via the phenotypic method and positive via PCR; still other isolates were found to be positive via the phenotypic method and negative via PCR. However, low template concentration, degraded DNA quality, or mutations at primer binding sites can all cause traditional PCR to produce false-negative findings. Furthermore, conventional PCR is unable to distinguish between genes situated on chromosomes and those placed on plasmids, nor does it reveal information on gene expression. Alternative processes, including as porin loss, efflux pump overexpression, and the overproduction of AmpC or extended-spectrum β-lactamases (ESBLs) in combination with decreased outer membrane permeability, can also lead to carbapenem resistance, as has been well documented in the literature [31]. When paired with β-lactamase synthesis, porin mutations or deletions decrease antibiotic inflow, which can result in high-level resistance even when carbapenemase genes are absent. In *B. cepacia*, which is renowned for having a wide range of resistance determinants and an inherently low outer membrane permeability, this is especially pertinent [21]. Furthermore, by actively extruding carbapenems and other β-lactams, efflux systems like the AcrAB-TolC pump and other resistance-nodulation-division (RND) family transporters may greatly increase resistance in *A. sobria* [17].

Using PCR (Polymerase Chain Reaction), the overall prevalence of carbapenemase genes was found to be 92.8% (52/56), with $bla_{KPC}$ accounting for 80.7% (42/52) and $bla_{GES}$ for 19.2% (10/52) of the total. The 42 *B. cepacia* isolates that tested positive for carbapenems included 38 $bla_{KPC}$ (n=38) and two $bla_{GES}$ (n=2); in contrast, four $bla_{KPC}$ (n=4) and eight $bla_{GES}$ (n=8) were present in the *A. sobria* isolates that tested positive for carbapenems. The present investigation found no positive results for the $bla_{IMP}$ gene of carbapenem-resistant *A. sobria* or *B. cepacia* isolates. In the present study, *B. cepacia* isolates carbapenems included 40 $bla_{KPC}$, in line with Al-Muhanna et al.'s 2020 study which found that the $bla_{KPC}$ gene yielded negative results for every *B. cepacia* isolate [4]. Haghighi and Goli, in 2022, found that two $bla_{GES}$ genes were detected in 83.72% of isolates among gram-negative bacteria like *Pseudomonas aeruginosa* [32]. Shanmugam and associates found that none of *Klebsiella pneumonia* isolates produced $bla_{GES}$. The transposable elements that surround the $bla_{KPC}$ genes, which encode $bla_{KPC}$, enable the gene to migrate back and forth between the bacterial chromosome and the transferable plasmid [33]. Difference in resistance to only one of the carbapenems and resistance to both imipenem and meropenem suggest the synthesis of carbapenemase, as this may indicate the existence of another form of resistance mechanism [34,35].

The present investigation found no positive results for the $bla_{IMP}$ gene of *A. sobria* or *B. cepacia* isolates, consistent with Al-Muhanna et al. who showed that *B. cepacia* isolates gave a negative $bla_{IMP}$ gene result [4]. According to recent research, mobile gene cassettes inserted into integrons carry two major groups of imported metallo-β−lactamases: VIM and IMP. With the exception of aztreonam, the majority of β-lactam antibiotics are hydrolyzed by metallo-β-lactamases; as a result, most β-lactam antibiotics, including the carbapenems, are ineffective against the many infections that produce these enzymes in high quantities [36,37]. A number of studies in Asian and European countries, the American countries including Brazil, Canada and the United States, and Australian countries have reported these isolates [38,39,40]. The availability of resistance mechanisms unrelated to carbapenemase synthesis, such as efflux pumps or porin loss, or the existence of additional carbapenemase genes (such as $bla_{NDM}$ and $bla_{VIM}$) not present in the current PCR panel could be the cause of these differences. Furthermore, false-negative PCR results could have been caused by mutations in primer-binding areas. Given these limitations, accurate carbapenem resistance detection requires the combination of phenotypic screening and molecular testing, especially in clinical settings where prompt and accurate monitoring of antibiotics is crucial. Ultimately, these resistance profiles carry significant clinical implications, often necessitating combination therapies or the use of second-line antibiotics, which may be more costly, associated with higher toxicity, or have reduced efficacy. Although direct associations with patient outcomes were outside the purview of this investigation, infections brought

on by multidrug-resistant *B. cepacia* and *A. sobria* are probably linked to longer hospital admissions, higher morbidity, and maybe worse clinical outcomes. The analysis did not include clinical outcome data, such as ICU admission, comorbidities, length of hospital stay, prior antibiotic exposure, or patient mortality. Although we acknowledge the significance of these host-related factors in comprehending the clinical consequences of antibiotic resistance, they were outside the purview of the current microbiological study. In order to more accurately evaluate the influence of resistance mechanisms on patient outcomes, we advise that future studies combine clinical and microbiological data.

## Conclusion

The main objectives of this work were to identify the resistance genes in isolates of *B. cepacia and A. sobria* molecularly and to detect antibiotic resistance phenotypically. According to this study, *B. cepacia* and *A. sobria* exhibit a high level of resistance to antibiotics, particularly to common β-lactam medications such as cefepime, piperacillin, and ceftriaxone. For infections involving these organisms, clinicians should exercise caution when selecting empirical antibiotic therapy, especially in high-risk patients. Instead, they ought to consider alternative treatments that are guided by susceptibility testing. As traditional phenotypic methods may not be precise enough to differentiate between various forms of carbapenemase, the prevalence of $bla_{KPC}$ and $bla_{GES}$ carbapenemase genes among the resistant isolates underscores the urgent need for routine molecular surveillance. This study shows that, from a diagnostic perspectives, the recA gene-based molecular approach provides an accurate and inexpensive alternative for the VITEK 2 system in recognizing *B. cepacia* complex members. PCR is still the best method for accurate carbapenemase gene detection because of its sensitivity and specificity. Whole-genome sequencing (WGS) of isolates that produce carbapenemase is recommended for future studies in order to determine their phylogenetic connects and potential for clonal propagation. Furthermore, more research should be done on *recA* nucleotide sequence analysis as a standardized technique for quickly identifying both new and old genomovars in the *B. cepacia* complex. This would improve patient treatment plans and infection control efforts by improving our understanding of each genomovar's clinical risks and pathogenic potential.

## Author contributions

**Conceptualization:** Mushtak T. S. Al-Ouqaili.

**Data curation:** Rawaa A. Hussein, Bushra A. Kanaan.

**Formal analysis:** Rawaa A. Hussein.

**Investigation:** Mushtak T. S. Al-Ouqaili, Rawaa A. Hussein.

**Methodology:** Bushra A. Kanaan, Ahmed T.S. Al-Neda.

**Project administration:** Mushtak T. S. Al-Ouqaili.

**Resources:** Ahmed T.S. Al-Neda.

**Supervision:** Mushtak T. S. Al-Ouqaili.

**Validation:** Rawaa A. Hussein.

**Visualization:** Ahmed T.S. Al-Neda.

**Writing – original draft:** Rawaa A. Hussein.

**Writing – review & editing:** Mushtak T. S. Al-Ouqaili.

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
