## [Decision Letter · Decision Letter 0]

2 May 2025

PONE-D-24-53922An investigation of carbapenemase-encoding genes in Burkholderia cepacia and Aeromonas sobria nosocomial infections among Iraqi patientsPLOS ONE

Dear Dr. Al-Ouqaili,

Thank you for submitting your manuscript to PLOS ONE. After careful consideration, we feel that it has merit but does not fully meet PLOS ONE’s publication criteria as it currently stands. Therefore, we invite you to submit a revised version of the manuscript that addresses the points raised during the review process.

We look forward to receiving your revised manuscript.

Kind regards,

Dwij Raj Bhatta, PhD

Academic Editor

PLOS ONE

Journal Requirements:

2. We note that your Data Availability Statement is currently as follows: All relevant data are within the manuscript and its Supporting Information files

5. We note that you have referenced (Hussein RA, Al-Ouqaili MTS, Majeed YH. Association between alcohol consumption, cigarette smoking, and Helicobacter pylori infection in Iraqi patients submitted to gastrointestinal endoscopy. J Emerg Med Trauma Acute Care [Internet]. 2022 Dec 22 [cited 2022 Dec 23];2022(6):12. Available from: https://www.qscience.com/content/journals/10.5339/jemtac.2022.aimco.12) which has currently not yet been accepted for publication. Please remove this from your References and amend this to state in the body of your manuscript: (ie “Bewick et al. [Unpublished]”) as detailed online in our guide for authors

Additional Editor Comments:

The manuscript provides important information on antibiotic sensitivity pattern of Burkholderia spp. and Aeromonas isolates from Iraq and resistance related marker genes. However authors need to revise manuscript as per reviewers comments! Adress all comments from reviewers!

Reviewers' comments:

Reviewer's Responses to Questions

Comments to the Author

1. Is the manuscript technically sound, and do the data support the conclusions?

Reviewer #1: Yes

Reviewer #2: Partly

2. Has the statistical analysis been performed appropriately and rigorously? 

Reviewer #1: No

Reviewer #2: No

3. Have the authors made all data underlying the findings in their manuscript fully available?

Reviewer #1: Yes

Reviewer #2: Yes

4. Is the manuscript presented in an intelligible fashion and written in standard English?

Reviewer #1: Yes

Reviewer #2: Yes

5. Review Comments to the Author

Reviewer #1: This study investigates the prevalence of antibiotic resistance, including carbapenem resistance, in Burkholderia cepacia and Aeromonas sobria isolates from clinical specimens in Iraq. It reported significant resistance rates to multiple antibiotics, with a predominance of blaKPC and blaGES carbapenemase producers, highlighting the need for rapid and accurate bacterial identification methods and better treatment strategies for multi-drug resistant infections.

The manuscript can be accepted only after incorporating following points:

Major comments

1.Please revise the title as Investigation of Carbapenemase-Encoding Genes in Burkholderia cepacia and Aeromonas sobria Isolates from Nosocomial Infections in Iraqi Patients

2.In abstract,

-The methods section is vague. Important details such as patient population, or sampling method are missing.

-There is some inconsistency in reporting numbers and percentages. For example You say “75 isolates of A. sobria and B. cepacia were assessed. A. sobria made up 16.6% (n=20), B. cepacia 45.8% (n=55)” — this totals 100% (75 isolates), but the calculation is different 20/75 = 26.6%, 55/75 = 73.3%. There is also a line that mentions “55/57 recA gene positive,” but earlier it says only 55 isolates of B. cepacia were included.

3.Introduction

- Several sentences repeat similar points (e.g., B. cepacia’s resistance and persistence are mentioned multiple times).

"Burkholderiaceae family (previously known as Pseudomonas cepacian)" is misleading. Pseudomonas cepacia was the former name of the species, not the family.

- Reference [2] on B. cepacia persistence and transmission seems dated or generalized; newer sources might better reflect current understanding of nosocomial spread.

- Consider briefly stating why A. sobria (less frequently reported than A. hydrophila) is the focus, especially when A. hydrophila is mentioned as more common.

-The study objective appears abruptly at the end. The authors are suggested for the transition into the aim more clearly by summarizing the knowledge gap.

4.Methodology Section:

-Although the study employs both VITEK 2 System and recA gene-based PCR for identification, more details on the specific conditions under which these methods were conducted would improve reproducibility. For example, did the VITEK 2 System settings or parameters vary from standard procedures? Similarly, the exact primers used for PCR should be included, along with details on the PCR conditions and any potential limitations of these methods.

-The description of carbapenemase testing through both phenotypic and genotypic approaches is valuable, but further clarification is needed on how results were interpreted in case of discrepancies between the two methods. This would help readers understand the rationale behind the final conclusions on resistance.

5.Discussion of Results:

-The discussion regarding the resistance profiles, especially with respect to B. cepacia and A. sobria, could be expanded to address the clinical implications of these findings more comprehensively. For example, how do these resistance patterns correlate with patient outcomes or impact therapeutic decisions? Also, how do the findings compare with other studies in the region or globally in terms of therapeutic options?

-The prevalence of carbapenemase-producing strains is discussed, but the manuscript could benefit from more in-depth analysis on the mechanisms of carbapenem resistance. For instance, are there any particular resistance mechanisms (e.g., porin loss, overproduction of β-lactamases) that could explain the observed high levels of resistance in B. cepacia and A. sobria? The manuscript suggests that other resistance mechanisms could be at play, but this could be explored in more detail.

6.Statistical Analysis:

While the study includes some breakdowns of resistance patterns (e.g., percentage of resistance), the manuscript lacks a detailed statistical analysis of the data. Including confidence intervals, significance testing (e.g., chi-square tests for categorical variables like gender and resistance patterns), and comparative analysis would improve the robustness of the findings.

Minor Comments and Typos:

1.Typo/Grammar:

o"No A. sobria was found in any of the cerebrospinal fluid specimens." → "No A. sobria isolates were found in any of the cerebrospinal fluid specimens."

o"This is consistent with Pineda-Reyes et al. (2024), who observed that A. sobri is linked to a variety of human infections..." → "A. sobria" (should match the species name format consistently).

oPlease be consistent in naming species, especially with the genus and species names like B. cepacia and A. sobria. Use italics throughout and ensure the first letter of the genus is capitalized.

oEnsure uniform citation formatting throughout the manuscript. There are a few inconsistencies in the way references are presented (e.g., reference numbering).

2.Figures/Tables:

oThe manuscript could benefit from including figures or tables that summarize the major findings, such as the resistance rates for each antibiotic tested, the types of specimens sampled, and a comparison of resistance patterns between B. cepacia and A. sobria. This would make the data more accessible and visually impactful.

3.Conclusion:

oThe conclusion is well-written but could be more focused on specific recommendations for future research and clinical practice. A few key research questions or clinical practice guidelines based on the findings would strengthen this section. For example, should clinicians be more cautious when using specific antibiotics for B. cepacia and A. sobria infections based on the resistance patterns observed?

oThere seems to be some inconsistency in the way data is reported. For example, you mention percentages in some cases but then refer to absolute numbers in others (e.g., "58.7% male" vs. "44 males"). It would be clearer and more standardized to report either both absolute numbers and percentages, or just one for consistency.

Reviewer #2: Review Report of the manuscript entitled An investigation of carbapenemase-encoding genes in Burkholderia cepacia and Aeromonas sobria nosocomial infections among Iraqi patients

This manuscript explores the prevalence of carbapenemase-encoding genes (blaKPC, blaGES, and blaIMP) among Burkholderia cepacia and Aeromonas sobria isolated from nosocomial infections in Iraq, using both phenotypic and molecular techniques. The topic is more relevant, especially in regions where antibiotic surveillance data are limited. The manuscript provides valuable local AMR data that may contribute to broader global antimicrobial resistance mapping. However, while the research question is relevant and the methods broadly appropriate, the manuscript further requires major revisions before it can be considered for publication.

1. Lack of clarity about the sampling method and inclusion/exclusion criteria. Please clarify.

2. Justification for selecting only three carbapenemase genes (blaKPC, blaGES, blaIMP) is insufficient—why you exclude blaNDM, blaOXA, blaVIM Please give the reasons?

3. Please give Controls for PCR and phenotypic testing (positive/negative) are not mentioned.

4. Only basic chi-square testing is used. A more robust analysis (e.g., multivariate regression) should be considered to correlate resistance genes with demographics or clinical variables.

5. Figures (e.g., gel electrophoresis) are low resolution and poorly labeled.

6. Tables need improved formatting; define all abbreviations and ensure legends are self-explanatory.

7. Sample size reporting is occasionally inconsistent (e.g., 75 vs 56 isolates).

8.Several results are repeated multiple times across the text, tables, and figures.

9.Lack of MIC50/MIC90 or ranges diminishes the quality of antibiotic resistance interpretation.

10.The discussion is largely descriptive and lacks critical evaluation of findings.

11. Clinical implications of findings are weakly addressed (e.g., impact on treatment policy or infection control practices).

12. Mentions “genomic sequencing” without this being part of the study.

13.Inconsistent use of scientific names (italicization of B. cepacia, A. sobria must be needed throughout the manuscript).

14. Abstract needs grammatical and syntactical revisions (e.g., “92.8%%” should be “92.8%”).

15 Some references are outdated, and certain statements require more recent supporting evidence.

16. In methodology: Study design (cross-sectional) and sampling methods are acceptable but

Was the sampling random or convenience-based? What were the inclusion/exclusion criteria?Ethical approval and consent are reported appropriately. Primer sequences and thermocycling protocols should be summarized more clearly in one table. Justification for excluding genes like blaNDM, blaOXA, or blaVIM is missing. The phenotypic test (double disk synergy) is correctly described but lacks proper validation or controls explanation.

17. There is confusion in sample size reporting: The abstract mentions 75 isolates, with 20 A. sobria and 55 B. cepacia, totaling 75, yet elsewhere “56 carbapenem-resistant isolates” are analyzed without clear criteria. Please give Justification??

18. Use of vague terms like “high”, “little resistance”, or “more accurate” should be replaced with quantified expressions.

19. Not all abbreviations are defined e.g., MEM, IMP are missing.

20.Some values are redundant across tables and figures.

21.The absence of sequencing data for PCR amplicons is a major limitation. Without sequencing, false-positive bands from non-specific amplification cannot be ruled out. Lack of controls (positive and negative) for each PCR gene detection is not mentioned.

22. According to CLSI guidelines, the double-disk synergy test (DDST) is not the gold standard for carbapenemase detection; tests like Carba NP or modified Hodge test should be used.

23. The study donot correlate clinical outcomes (e.g., mortality, hospital stay) with resistance profiles. Host factors, such as patient comorbidities, prior antibiotic exposure, and ICU admission, are not considered in the study. Please consdider for these factors.

24.The authors state adherence to the “Helsinki Declaration of 1979,” which is outdated; the 2013 version is recent Helsinki Declaration.

Final Recommendation: Major Revisions

6. PLOS authors have the option to publish the peer review history of their article (what does this mean? ). If published, this will include your full peer review and any attached files.

**Do you want your identity to be public for this peer review?** For information about this choice, including consent withdrawal, please see our Privacy Policy .

Reviewer #1: No

Reviewer #2: **Yes: ** Komal Raj Rijal

---

## [Author Response · Author response to Decision Letter 1]

12 May 2025

Dear Dr. Dwij Raj Bhatta

Academic Editor of PLOS ONE

Nice Greetings

Subject : Submission of revised manuscript entitled "An investigation of carbapenemase-encoding genes in Burkholderia cepacia and Aeromonas sobria nosocomial infections among Iraqi patients". (PONE-D-24-53922)

Thank you for your email of 2 May, 2025 enclosing the editor and reviewers ʼcomments. We also greatly appreciate the reviewers for their complimentary comments and suggestions. We have carefully reviewed the comments and have revised the manuscript accordingly. Our point to point responses are given below. (The reviewer’s comments are in italics). Changes to the manuscript are indicated in yellow highlighted sentences/words. We hope that you find our responses satisfactory and that the manuscript is now acceptable for publication. Anyway, we should be grateful if you let us know about our further changes required.

Yours sincerely,

Corresponding author: Mushtak T. S. Al-Ouqaili

Department of Microbiology, College of Medicine, University of Anbar, Al-Anbar Governorate, Ramadi, Iraq Email: ph.dr.mushtak_72@uoanbar.edu.iq

Tel: +9647830014212

The followings are point-by-point responses :

Response to Editor

We note that you have referenced (Hussein RA, Al-Ouqaili MTS, Majeed YH. Association between alcohol consumption, cigarette smoking, and Helicobacter pylori infection in Iraqi patients submitted to gastrointestinal endoscopy. J Emerg Med Trauma Acute Care [Internet]. 2022 Dec 22 [cited 2022 Dec 23];2022(6):12. Available from: https://www.qscience.com/content/journals/10.5339/jemtac.2022.aimco.12) which has currently not yet been accepted for publication. Please remove this from your References and amend this to state in the body of your manuscript: (i.e “Bewick et al. [Unpublished]”) as detailed online in our guide for authors

Response to Editor:

Thank you for highlighting this important point. We have carefully reviewed and revised the manuscript to ensure uniform citation formatting throughout. All reference numbers have been checked for accuracy and consistency, and the citation style has been standardized in accordance with the journal’s guidelines.

This reference was removed. “Hussein RA, Al-Ouqaili MTS, Majeed YH. Association between alcohol consumption, cigarette smoking, and Helicobacter pylori infection in Iraqi patients submitted to gastrointestinal endoscopy. J Emerg Med Trauma Acute Care [Internet]. 2022 Dec 22 [cited 2022 Dec 23];2022(6):12. Available from: https://www.qscience.com/content/journals/10.5339/jemtac.2022.aimco.12”

Responses to Reviewer's Comments:

Major comments

1. Please revise the title as Investigation of Carbapenemase-Encoding Genes in Burkholderia cepacia and Aeromonas sobria Isolates from Nosocomial Infections in Iraqi Patients

Response to Reviewer:

I appreciate your insightful comments. Following your suggestion, we have revised the title to " Investigation of Carbapenemase-Encoding Genes in Burkholderia cepacia and Aeromonas sobria Isolates from Nosocomial Infections in Iraqi Patients." This updated title better reflects the content and focus of our study. Thank you for your recommendations on making the manuscript more accessible.

2. In abstract,

-The methods section is vague. Important details such as patient population, or sampling method are missing.

Response to Reviewer:

Thank you for your valuable feedback. In response to your comments regarding the clarity of the Methods section, we have revised it to include additional details about specimen sources, identification procedures, antibiotic susceptibility testing, and molecular analyses. We appreciate your helpful suggestions and thank you for your contribution to improving the manuscript.

"A total of 120 clinical specimens from patients with nosocomial infections were collected randomly. Ear swabs, urine, burns, wounds, and cerebrospinal fluid samples were all cultured on selective media. Using biochemical assays, Burkholderia cepacia and Aeromonas sobria were identified, and the VITEK-2 system confirmed their identities. VITEK-2 was utilized to perform antibiotic susceptibility testing in accordance with CLSI standards. The meropenem-EDTA combination disc test was used to phenotypically screen for the production of metallo-β-lactamase (MBL). The recA gene in B. cepacia and the genes encoding carbapenemase in both species were found using PCR tests. "

There is some inconsistency in reporting numbers and percentages. For example You say “75 isolates of A. sobria and B. cepacia were assessed. A. sobria made up 16.6% (n=20), B. cepacia 45.8% (n=55)” — this totals 100% (75 isolates), but the calculation is different 20/75 = 26.6%, 55/75 = 73.3%.

There is also a line that mentions “55/57 recA gene positive,” but earlier it says only 55 isolates of B. cepacia were included.

Response to Reviewer:

Thank you for pointing out the inconsistency in the reporting of percentages. We have corrected the statement to reflect the accurate proportions based on the total of 75 isolates, with A. sobria comprising 26.6% (n=20) and B. cepacia comprising 73.3% (n=55).

Response to Reviewer

Thank you for highlighting this difference.. We have clarified that only 55 isolates tested positive for the recA gene, although 57 isolates were originally diagnosed as B. cepacia using the VITEK 2 technology. The VITEK 2 system may have misidentified the two negative isolates, which can happen with closely similar non-fermenting bacteria.

The text has been updated to reflect that all 55 isolates that the VITEK 2 system identified as B. cepacia also tested positive for the recA gene. This supports the reliability of the identification and validates their classification within the B. cepacia complex. This discovery is now appropriately reflected in the manuscript's updated sentence. We appreciate your careful review and constructive feedback.

1. Introduction

- Several sentences repeat similar points (e.g., B. cepacia’s resistance and persistence are mentioned multiple times).

Thank you for your insightful observation regarding the redundancy in the background section. We have revised the text to eliminate any repetitions, particularly concerning the persistence and resistance mechanisms of Burkholderia cepacia. While maintaining all the essential background information to provide context for the study, we have streamlined the section for improved clarity and scientific flow. We appreciate your constructive criticism, which has enhanced the overall quality of the manuscript.

“The bacterial isolates of B. cepacia frequently show intrinsic resistance to several antibiotics or drugs classes, making infections difficult to treat. This resistance, which includes decreased susceptibility to substances like ticarcillin, cephalosporins, phosphonic acid antibiotics, polymyxins, and aminoglycosides, develops spontaneously and does not require gene acquisition or mutation“

"Burkholderiaceae family (previously known as Pseudomonas cepacian)" is misleading. Pseudomonas cepacia was the former name of the species, not the family.

Response to Reviewer

Thank you for highlighting the taxonomic inaccuracy. We have revised the text to clarify that Pseudomonas cepacia was the former name of the species, not of the Burkholderiaceae family. The corrected sentence now accurately reflects both the historical classification and the current taxonomy of Burkholderia cepacia. We appreciate your thorough review and valuable feedback.

“Burkholderia cepacia is a Gram-negative, rod-shaped, motile bacterium belonging to the Burkholderiaceae family (formerly known as Pseudomonas cepacia).”

Reference [2] on B. cepacia persistence and transmission seems dated or generalized; newer sources might better reflect current understanding of nosocomial spread.

Response to Reviewer:

Thank you for your insightful comment regarding the need for a more current reference on Burkholderia cepacia's persistence and transmission. In response, we have updated the manuscript to include a recent and more relevant source: Silva-Santana G, Sales FLS, Aguiar AR, Brandão MLL. Pharmaceutical Contamination by Biofilms Formed of the Burkholderia cepacia Complex: Public Health Risks. Processes. 2025;13(5):1270. https://doi.org/10.3390/pr13051270

Consider briefly stating why A. sobria (less frequently reported than A. hydrophila) is the focus, especially when A. hydrophila is mentioned as more common.

Response to Reviewer:

Thank you for your insightful observation. While A. hydrophila is indeed more frequently reported in clinical settings, our focus on A. sobria arises from its emerging role as a pathogen in nosocomial infections, particularly in the region where our study was conducted. Recent isolates from patients in our cohort have shown a significant presence of A. sobria, which warrants further investigation into its resistance patterns and potential public health implications. In the revised manuscript, we have briefly clarified this rationale to provide better context for its inclusion.

“The current study focuses on the prevalence of A. sobria, highlighting its increasing clinical significance and justifying its inclusion in this analysis, despite the fact that A. hydrophila is more commonly reported in clinical infections.”

The study objective appears abruptly at the end. The authors are suggested for the transition into the aim more clearly by summarizing the knowledge gap.

Response to Reviewer:

Thank you for your valuable suggestion. We have revised the manuscript to improve the transition into the study's objective by first summarizing the knowledge gap. This enhancement provides clearer context and emphasizes the importance of investigating carbapenemase genes and antibiotic resistance patterns in B. cepacia and A. sobria. We appreciate your feedback, as it has significantly improved the clarity of the manuscript.

“Although A. sobria and B. cepacia are becoming more widely recognized as important nosocomial infection pathogens, little is known about the molecular resistance mechanisms of these bacteria, especially in relation to carbapenemase genes. More research on the resistance characteristics of multidrug-resistant bacteria is essential due to their rising prevalence. The aim of this study was to investigate the occurrence of carbapenemase genes and patterns of antibiotic resistance in isolates of B. cepacia and A. sobria.”

4- Methodology Section:

- Although the study employs both VITEK 2 System and recA gene-based PCR for identification, more details on the specific conditions under which these methods were conducted would improve reproducibility. For example, did the VITEK 2 System settings or parameters vary from standard procedures? Similarly, the exact primers used for PCR should be included, along with details on the PCR conditions and any potential limitations of these methods.

Response to Reviewer:

We appreciate the reviewer’s valuable feedback regarding the need for increased methodological clarity. In response, we have revised the Methods section to include more detailed information on the identification procedures. Specifically, we now specify that the initial identification was based on traditional bacteriological methods, which included Gram staining, oxidase testing, motility testing, and growth on selective media. Final confirmation was performed using the VITEK® 2 Compact B System (BioMérieux, France) with VITEK® 2 GN ID cards, in accordance with the manufacturer’s guidelines. No modifications or deviations from the standard operating procedure were made. The instrument automatically processed bacterial suspensions prepared according to BioMérieux’s instructions, ensuring consistency and reproducibility across all tested isolates. These enhancements aim to improve the reproducibility and transparency of the identification process.

“Gram-negative isolates suspected of being A. sobria or B. cepacia were initially identified using normal bacteriological techniques. According to MacFaddin's (2000) procedures, these included Gram staining, oxidase and motility tests, and growth assessment on selective medium such MacConkey agar, blood agar, and mannitol salt agar. Morphological and microscopic features were also assessed. The VITEK® 2 Compact B System (BioMérieux, France) with VITEK® 2 GN ID cards was used to confirm the final species-level identification in accordance with the manufacturer's instructions.”

Response to Reviewer:

We appreciate the reviewer's insightful recommendation. In response, we would like to clarify that the manuscript already includes detailed descriptions of the PCR protocols used for detecting both the recA gene and the carbapenemase genes. These descriptions specify the primer sequences, product sizes, annealing temperatures, and the exact thermal cycling conditions for each gene target, as well as the reagents and volumes used in the reaction mixtures (see Tables 1 and 2). This level of detail was included to ensure transparency and enhance the reproducibility of our methods. Additionally, we have cited relevant references and indicated the use of validated primers. We are happy to provide further information on these topics if needed.

Additionally, a brief assessment of these strategies' limitations has been added. However, low template concentration, degraded DNA quality, or mutations at primer binding sites can all cause traditional PCR to produce false-negative findings. Furthermore, conventional PCR is unable to distinguish between genes situated on chromosomes and those placed on plasmids, nor does it reveal information on gene expression.

Brisse and colleagues discovered that, despite the fact that B. gladioli is not listed in the VITEK2 database (and so should have a higher probability of misidentification), only one isolate of the bacteria was recognized by the VITEK 2 apparatus as "B. cepacia or B. pseudomalleii." However, as other isolates of these species or other non-fermenters like Achromobacter or Alcaligenes may be mistakenly recognized as B. cepacia, it is always an excellent choice to use a molecular approach to validate the B. cepacia identification [25]. The recA gene has been widely used in bacterial systematics and has proven to be highly helpful in identifying species of the B. cepacia complex. By using phylogenetic analysis of sequence variance within the gene, it is possible to distinguish between the nine present species of the B. cepacia complex

PCR techniques for recA gene detection in B. cepacia isolates

Using the recA gene of the Bcc complex in clinical samples and BCR1 and BCR2 primers, a standard PCR technique was used to directly molecularly identify the Bcc complex in all 75 extracted DNA samples (Table 1). A 25 μl reaction system comprising 12.5 μl of GoTaq® Green Master Mix (2X) (Promega, USA), 1 μl of each forward and reverse primer, 7.5 μl of nuclease-free water, and 3 μl of extracted DNA template was used for the PCR experiments. For the recA gene, the PCR reaction protocol was set up as follows: 30 cycles of 45 seconds at 94 O C, 45 seconds at 56 O C, and 90 seconds at 72 O C were followed by one cycle of 94 °C for two minutes. At 72°C, the last extension step was set for a single 7-minute cycle. Using a molecular size measurement of one hundred base pairs (from Bioneer, Korea), the PCR products were seen on a 1.5% agarose gel stained with red safe nucleic acid staining (Intron, Korea) [17].

Molecular methods for identification of carbapenemase genes

12.5 μl of GoTaq® Green Master Mix (2X), 1 μl of forward and reverse primers (Macrogen, Korea), 3 μl of bacterial DNA, 0.5 μl of MgCl2, and 7 μl of double distilled and deionized water made up a total volume of 25 μl. The following conditions for PCR cycling were employed: 30 cycles of one minute each at 95oC, one minute at 50 or 55oC, one minute at 72oC, and ten minutes at 72oC (as indicated by Table 2). The PCR product was analyzed using a 100 bp molecular size marker gel and detected using Red Safe Nucleic Acid Staining [16] (Intron, Korea).

Gene Name Primer Sequence (5′–3′) Product Size (bp) Annealing Temp (°C) Reference

B. cepacia complex recA BCR1 TGACCGCCGAGAAGAGCAA 104

---

## [Decision Letter · Decision Letter 1]

13 Jul 2025

PONE-D-24-53922R1Investigation of Carbapenemase-Encoding Genes in Burkholderia cepacia and Aeromonas sobria Isolates from Nosocomial Infections in Iraqi PatientsPLOS ONE

Dear Dr. Al-Ouqaili,

Thank you for submitting your manuscript to PLOS ONE. After careful consideration, we feel that it has merit but does not fully meet PLOS ONE’s publication criteria as it currently stands. Therefore, we invite you to submit a revised version of the manuscript that addresses the points raised during the review process.

We look forward to receiving your revised manuscript.

Kind regards,

Hideo Kato

Academic Editor

PLOS ONE

Reviewers' comments:

Reviewer's Responses to Questions

**Comments to the Author**

1. If the authors have adequately addressed your comments raised in a previous round of review and you feel that this manuscript is now acceptable for publication, you may indicate that here to bypass the “Comments to the Author” section, enter your conflict of interest statement in the “Confidential to Editor” section, and submit your "Accept" recommendation.

Reviewer #3: All comments have been addressed

Reviewer #4: All comments have been addressed

Reviewer #5: (No Response)

2. Is the manuscript technically sound, and do the data support the conclusions?

Reviewer #3: Yes

Reviewer #4: Partly

Reviewer #5: Yes

3. Has the statistical analysis been performed appropriately and rigorously? 

Reviewer #3: Yes

Reviewer #4: N/A

Reviewer #5: Yes

4. Have the authors made all data underlying the findings in their manuscript fully available?

Reviewer #3: Yes

Reviewer #4: Yes

Reviewer #5: Yes

5. Is the manuscript presented in an intelligible fashion and written in standard English?

Reviewer #3: Yes

Reviewer #4: Yes

Reviewer #5: No

6. Review Comments to the Author

Reviewer #3: The authors have done well in addressing the comments raised and the work is technically sound, it is a commendable work which can make significant impact in the field of medical and pharmaceutical microbiology

Reviewer #4: Authors have made a significant changes in the manuscripts. However, before final remarks, some are needs to be improved for better clarification.

Abstract:

1. You have mentioned that you collected 120 samples, and assessed 75 isolates for further identification. did you mean that 75 isolates were positive in culture out of 120 samples? If so then add it and clarify it or if not so then mention how did you choose 75 isolates?

2. You have mentioned "The study isolates showed

highest antimicrobial resistance to piperacillin, cefepime, ceftriaxone (100%), ceftazidime

(97.3%), and lowest antimicrobial resistance to imipenem (36%)". it is better to add the resistance percentage of all antibiotics for scientific clarity.

3. In a few areas of the whole paper, you have mentioned a data like only number. For as an example " The 42 B. cepacia isolates that tested positive for carbapenem

resistance were constituted of 38 blaKPC (n = 38) and two blaGES (n = 2); in contrast, four blaKPC

(n = 4) and eight blaGES (n = 8) were present in the A. sobria isolates that tested positive for

carbapenems resistance" However, it is better to add percentage as well with the number.

4. You have added "None of isolates studied tested positive for the blaIMP gene. The recent study concluded that recA gene identification was more sensitive and specific technique for detection B. cepacia complex isolates. There was a notable predominance of blaKPC and blaGES carbapenemase producers among the isolates under investigation. The blaIMP gene was not found in any of the research isolates". Please remove the last two line from " There was a notable predominance of blaKPC and blaGES carbapenemase producers among the isolates under investigation. The blaIMP gene was not found in any of the research isolates" as you mentioned earlier. Instead of this, add a remarks which steps should be considered to reduce this the threat during treatment by a clinician or something like that.

Materials and Methods

5. you have mentioned " Many of the samples were cultivated on culture media such as MacConkey

agar, Blood agar, and Mannitol agar (Oxoid, UK), as obtained from CSF, urine, ear swabs, burns,

wounds and diabetic foot ulcer". But it's not clear how many have you done and add the result of this in the result section. Otherwise its too vague to understand.

6. You have described that you have identified these isolates "According to MacFaddin's (2000) procedures" but did not cite this reference, please add the reference for reader to find clearly.

7. You have done Antibiotic Susceptibility by CLSI guildlike but did not mention the year for CLSI, as it is changing day by day or CLSI is publishing new reference to use for researcher.

8. You have mentioned "A total of 56 strains resistant to at least one carbapenem were investigated" but ther number you have found did not included in previous section as it is in result section. So it is better to the number in only result section and escribe it like " The isolates that was resistant to at least one carbapenem were investigated for MBL production.

9. In DNA extraction and PCR sectionS, you have mentioned " the entire DNA of 75 B. cepacia and 20

A. sobria isolates was extracted" is it 75 B. cepacia or it would be 55?? As earlier you have added in the asbtract that you found 55 B. cepacia!!!!. It is also better to skip the number in methods, just add in result section like, "in molecular detecction of 55 B. cepacia and 20 A. sobria, we found...........

10. In statistical methods, you did not mention how did you calculate 95% CI. Please add.

Results:

11. You have outlined " Among the 75 isolates assessed 20 (26.6%) were A. sobria and 55 (73.3%) were B.

cepacia" it is better to say like "Among the 75 culture positive isolates assessed in this study, 20 (26.6%) were A. sobria and 55 (73.3%) were B. cepacia. or add from you have selected or found 75 isolates.

12. You have added "The average age of the study participants, who ranged in age from 10 to 70,

was 40.13 ± 22.5 years" but did not included in table 2. Please add for all the categories for better understanding.

13. Pleas add percentage here for each antibiotics "The isolates showed resistance to cefepime, piperacillin, and ceftriaxone (100%),ceftazidime (97.3%), and imipenem (36%)"

14. In figure 2, it seems L3 is negative or absence of positive bands but you did not added this information the the figure legend.

15. In the "Phenotypic and Genotypic Validation" section you described : There was no gene present in two

isolates that were resistant to carbapenem. None of the isolates contained the blaIMP gene, as can

be seen in Figures 4, 5. There was no gene present in the two isolates resistant to carbapenem. It seems like there is repetition. Please clarify.

16. In a previous question "The prevalence of carbapenemase-producing strains is discussed, but the manuscript

could benefit from more in-depth analysis on the mechanisms of carbapenem

resistance. For instance, are there any particular resistance mechanisms (e.g., porin

loss, overproduction of β-lactamases) that could explain the observed high levels of

resistance in B. cepacia and A. sobria? The manuscript suggests that other resistance

mechanisms could be at play, but this could be explored in more detail", you have added the description in the result section and discussion both. But it is better to add only in discussion section to make the result section more clear and concise.

17.The abbreviation of PCR should only be "*PCR: Polymerase Chain Reaction"

18. Result section seems to large. It is better to make it concise by adding only the findings from your study and the information like your theoritical observation would be kept in discussion section only.

19. Please add reference for below information in discussion parrt "Carbapenemases are enzymes that have various different hydrolytic profiles; all are categorized as �-lactamases. All the �-lactam antibiotics, including the penicillins,

cephalosporins, monobactams, and the carbapenems, are acted upon by these enzymes. Due to

carbapenemase activity, a majority of �-lactam drugs can become inactive against the severe

infections that result from bacteria that produce such �-lactamases. Enzyme superfamilies’

comprise carbapenemases from the A, B, and D classes of �-lactamases. Zinc is present in the

active region of class B enzymes, which are metallo-�-lactamases; class A and D enzymes, on

the other hand, hydrolyze serine. The Guiana-Extended-Spectrum and Klebsiella pneumoniae

18carbapenemase families are amongst the Class A carbapenemases

20. If possible reduce the discussion and keep only relevant information though it is well described. But reducing will enhance the clarify for better understanding.

21. In conclusion, you have added "The main objectives of this work were to identify the resistance genes in isolates

of B. cepacia and A. sobria molecularly and to detect antibiotic resistance phenotypically" in the middle part, but it would be better to add in the first line then followed by your major findings and conclusion remarks.

22. You also added "The analysis did not include clinical outcome data, such as ICU admission, comorbidities, length of hospital stay, prior antibiotic exposure, or patient mortality. Although we acknowledge the significance of these host-related factors in comprehending the clinical consequences of antibiotic resistance, they were outside the purview of the current microbiological study" in the conclusion but it would be better to add in the end part of the discussion to illustrate your limitation and for better understanding of your conclusion remarks.

Reviewer #5: Thank you for your efforts in revising the manuscript. While I acknowledge that some reviewer comments have been addressed, several critical issues remain that require your attention to improve the manuscript’s quality and clarity.

General Comments:

Manuscript Formatting: The absence of line numbers made it difficult to conduct an efficient review. In future resubmissions, please ensure that all revisions are clearly indicated and line numbers are included.

Language and Style: The revised manuscript still includes numerous typographical errors and lacks an academic tone in several sections. It is strongly recommended to have the manuscript professionally edited or reviewed by a fluent English speaker.

Discussion Section: The beginning of the discussion section closely mirrors the results, without providing critical analysis. The discussion should succinctly highlight key findings, contextualize them with relevant literature, and explain their broader implications.

Specific Comments:

Introduction:

Line 12: The phrase “The bacterial isolates of B. cepacian” should be revised to “B. cepacian isolates” for conciseness and grammatical accuracy.

Page 4, Line 6: Please correct the phrase “by using the random sampling procedure..” by removing the extra period.

Page 4, Lines 8–20: The inclusion of general precautions for handling B. cepacian and A. sobria seems misplaced in the methodology section. The methods should focus on detailing the specific procedures you followed, not general laboratory guidelines.

Discussion:

Page 20, Lines 1–8: The current text—“In ultimately, these resistance profiles have significant clinical implications. They need the use of combination or second-line treatments, which could be more expensive, hazardous, or less effective.”—should be revised for clarity and academic tone. Suggested revision:

"Ultimately, these resistance profiles carry significant clinical implications, often necessitating combination therapies or the use of second-line antibiotics, which may be more costly, associated with higher toxicity, or have reduced efficacy."

References: Several citations are outdated (e.g., references 3, 5, 17, 18, 25). It is advised to incorporate recent and relevant literature to strengthen the scientific foundation of the manuscript.

I hope these comments are helpful in guiding further improvements. I look forward to reviewing a revised version that addresses these concerns more thoroughly.

Best regards

7. PLOS authors have the option to publish the peer review history of their article (what does this mean? ). If published, this will include your full peer review and any attached files.

**Do you want your identity to be public for this peer review?** For information about this choice, including consent withdrawal, please see our Privacy Policy .

Reviewer #3: **Yes: ** Dr. Salim Faruk Bashir

Reviewer #4: No

Reviewer #5: No

---

## [Author Response · Author response to Decision Letter 2]

17 Jul 2025

Dear Dr. Hideo Kato

Academic Editor of PLOS ONE

Subject: Submission of revised manuscript entitled "Investigation of Carbapenemase-Encoding Genes in Burkholderia cepacia and Aeromonas sobria Isolates from Nosocomial Infections in Iraqi Patients". (PONE-D-24-53922R1)

Thank you for your email of 14 July 2025, enclosing the editor and reviewers' comments. We also greatly appreciate the reviewers for their complimentary comments and suggestions. We have carefully reviewed the comments and have revised the manuscript accordingly. Our point-to-point responses are given below. (The reviewer's comments are in italics.) Changes to the manuscript are indicated in yellow highlighted sentences/words. We hope that you find our responses satisfactory and that the manuscript is now acceptable for publication. Anyway, we would appreciate it if you could let us know about any further changes that may be required.

Yours sincerely,

Corresponding author: Mushtak T. S. Al-Ouqaili

Department of Microbiology, College of Medicine, University of Anbar, Al-Anbar Governorate, Ramadi, Iraq Email: ph.dr.mushtak_72@uoanbar.edu.iq

Tel: +9647830014212

The following are point-by-point responses :

Responses to Reviewer's Comments:

Reviewer #3: The authors have done well in addressing the comments raised, and the work is technically sound. It is a commendable work that can make a significant impact in the field of medical and pharmaceutical microbiology.

Reviewer #4: The Authors have made significant changes in the manuscript. However, before final remarks, some need to be improved for better clarification.

1. You have mentioned that you collected 120 samples and assessed 75 isolates for further identification. Did you mean that 75 isolates were positive in culture out of 120 samples? If so, then add it and clarify it, or if not, then mention how you chose 75 isolates.

Response to Reviewer Comment:

Thank you for your observation. We appreciate the opportunity to clarify this point.

In our study, a total of 120 clinical specimens were collected from patients with nosocomial infections. These specimens included ear swabs, urine, burn, wound, and cerebrospinal fluid samples. Out of these 120 samples, 75 yielded positive cultures for either Burkholderia cepacia or Aeromonas sobria. These 75 culture-positive isolates were then subjected to further identification and antimicrobial resistance testing.

We have revised the relevant section of the manuscript to clarify this point as follows:

Randomly, 120 clinical specimens have been collected from patients with nosocomial infections. Selective media were used to culture ear swabs, urine, burns, wounds, and cerebrospinal fluids. According to biochemical tests and the VITEK-2 system, 75% of these demonstrated positive growth with B. cepacia and A. sobria.

2. You have mentioned "The study isolates showed

The highest antimicrobial resistance was to piperacillin, cefepime, ceftriaxone (100%), ceftazidime(97.3%), and the lowest antimicrobial resistance to imipenem (36%). It is better to add the resistance percentage of all antibiotics for scientific clarity.

Response to Reviewer Comment:

Thank you for your valuable suggestion. We agree that providing the resistance percentages for all antibiotics tested enhances the scientific clarity and completeness of the results. Accordingly, we have revised the relevant section of the manuscript to include detailed resistance data for each antibiotic.

The updated section now reads:

Piperacillin, cefepime, and ceftriaxone showed antimicrobial resistance of 100%, followed by ceftazidime (97.3 %), cefazolin (96 %), and piperacillin/ tazobactam (94.6 %). Intermediate resistance was reported with aztreonam (61.3%), meropenem (49.3%), trimethoprim-sulfamethoxazole (49.3%), gentamicin (46.6%), levofloxacin (44%), and ciprofloxacin (44%). It is important to note that minocycline (40%), amikacin (40%), imipenem (36%), and tigecycline (34.6%) had the lowest resistance rates, hence their relatively higher efficacy against the tested isolates.

3. In a few areas of the whole paper, you have mentioned data like only numbers. For example, " The 42 B. cepacia isolates that tested positive for carbapenem

resistance was constituted of 38 blaKPC (n = 38) and two blaGES (n = 2); in contrast, four blaKPC(n = 4) and eight blaGES (n = 8) were present in the A. sobria isolates that tested positive for carbapenem resistance.

However, it is better to add a percentage as well with the number.

Response to Reviewer Comment:

Thank you for your valuable observation. In response, we have revised the relevant sections throughout the manuscript to include percentages alongside absolute numbers for greater clarity and scientific accuracy. For example, the section discussing the distribution of carbapenemase genes among carbapenem-resistant isolates now reads:

Specifically, 38 (90.51%) of the 42 (76.36%) B. cepacia isolates that were positive in carbapenem resistance carried the blaKPC gene, 2 (4.81%) isolates carried blaGES, and 2 (4.81%) had no detectable carbapenemase gene. In the case of the 14 A. sobria carbapenem-resistant isolates, there were 4 isolates (28.6%) that had blaKPC, 8 isolates (57.1%) that had blaGES, and 2 isolates (14.3%) that did not have any carbapenemase genes.

4. You have added "None of the isolates studied tested positive for the blaIMP gene. The recent study concluded that recA gene identification was a more sensitive and specific technique for the detection of B. cepacia complex isolates. There was a notable predominance of blaKPC and blaGES carbapenemase producers among the isolates under investigation. The blaIMP gene was not found in any of the research isolates. Please remove the last two lines from " There was a notable predominance of blaKPC and blaGES carbapenemase producers among the isolates under investigation. The blaIMP gene was not found in any of the research isolates," as you mentioned earlier. Instead of this, add a remark on which steps should be considered to reduce the threat during treatment by a clinician, or something like that.

Response to Reviewer Comment:

Thank you for your insightful suggestion. We have removed the last two sentences as requested and replaced them with remarks emphasizing clinical considerations to mitigate the threat posed by carbapenem-resistant B. cepacia and A. sobria isolates. The revised text now highlights important steps clinicians can take during treatment to address this challenge.

The updated section now reads:

Since the prevalence of carbapenemase producers is high, careful infection control measures, rapid diagnostics, and antimicrobial stewardship must be implemented by clinicians. It is necessary that combination therapy be guided and early detectable to ensure better outcomes and restrict resistance.

Materials and Methods

5. You have mentioned " Many of the samples were cultivated on culture media such as MacConkey agar, Blood agar, and Mannitol agar (Oxoid, UK), as obtained from CSF, urine, ear swabs, burns, wounds, and diabetic foot ulcer". But it's not clear how many you have done, and add the result of this to the result section. Otherwise, it's too vague to understand.

Response to Reviewer Comment:

Thank you for pointing this out. To address the concern, we have clarified the total number of clinical specimens (n = 120), specified the types of samples, and added the relevant distribution details in the Results section. This improves transparency regarding sample sources and their respective cultural outcomes.

Revised Methods (clarification added):

There were a total of 120 clinical specimens from randomly selected subjects with suspected nosocomial infections. Samples were cultured on MacConkey agar, blood agar, and mannitol salt agar (Oxoid, UK).

Revised Results (addition of sample distribution):

Patients with suspected nosocomial infections provided a total of 120 clinical specimens, which were chosen at random. 75 (62.5%) of the 120 specimens showed positive growth for A. sobria and B. cepacia. Of the isolates that tested positive, 20 (26.6%) were found to be A. sobria and 55 (73.3%) to be B. cepacia. Wound swabs accounted for the majority of isolates (45.3%), followed by urine samples (33.3%), diabetic foot ulcers (9.3%), and other clinical sources (Table 3).

6. You have described that you have identified these isolates, "According to MacFaddin's (2000) procedures," but did not cite this reference. Please add the reference for the reader to find clearly.

Response to Reviewer Comment:

Thank you for your observation. We have now included the full citation for MacFaddin (2000) in the reference list to support the identification procedures described in the methodology, as requested.

Reference List Addition:

13. MacFaddin JF. Biochemical tests for the identification of medical bacteria. 3rd ed. USA: Lippincott Williams & Wilkins; 2000.

7. You have done Antibiotic Susceptibility by the CLSI guideline, but did not mention the year for CLSI, as it is changing day by day, or CLSI is publishing new references to use for researchers.

Response to Reviewer Comment:

Thank you for your important observation. We have now specified the CLSI guideline year (2024) in the Methods section and added the corresponding reference to ensure clarity and alignment with current standards.

Updated In-Text (Methods section):

According to the criteria defined by the Clinical and Laboratory Standards Institute (CLSI, 2024), the antibiotic susceptibilities of B. cepacia and A. sobria isolates were tested using the VITEK-2 System (BioMérieux, Marcyl'Étoile, France)

9. In the DNA extraction and PCR sections, you have mentioned " the entire DNA of 75 B. cepacia and 20 A. sobria isolates was extracted." Is it 75 B. cepacia or 55?? As earlier, you have added in the abstract that you found 55 B. cepacia!!!!. It is also better to skip the number in methods, just add in the result section like, "in molecular detection of 55 B. cepacia and 20 A. sobria, we found...........

Response to Reviewer Comment:

Thank you for catching the inconsistency. You are correct—the number of B. cepacia isolates is 55, not 75, as stated earlier. We have removed the specific isolate numbers from the Methods section to avoid confusion and instead presented the exact figures in the Results section as suggested.

Revised Methods Section:

B. cepacia and A. sobria isolates were grown overnight in brain heart infusion broth under completely aseptic conditions to obtain the purified single colonies. Genomic DNA was extracted after incubation with DNA extraction kits of commercial origin (Sacace, Italy), in accordance with the manufacturer-provided guidelines.

Revised Result Section:

The molecular detection analysis was performed by testing 55 (73.33%) B. cepacia and 20 (26.66%) A. sobria isolates' DNA in search of the possession of carbapenemase genes by PCR.

10. In statistical methods, you did not mention how you calculate the 95% CI. Please add.

Response to Reviewer:

Thank you for your valuable observation. We have now clarified the method used to calculate the 95% confidence intervals in the revised manuscript. Specifically, we have added the following sentence to the Statistical Methods section:

The Wilson score method was used to calculate the 95% CI of the proportion

Results:

11. You have outlined " Among the 75 isolates assessed 20 (26.6%) were A. sobria and 55 (73.3%) were B. cepacia" it is better to say like "Among the 75 culture positive isolates assessed in this study, 20 (26.6%) were A. sobria and 55 (73.3%) were B. cepacia. or add from you have selected or found 75

Response to Reviewer:

Thank you for your suggestion. We agree that additional clarification is needed. We have revised the sentence for clarity and now specify that the 75 isolates were culture-positive. The updated sentence in the results section reads as follows:

Among the 75 culture-positive isolates evaluated in this study, 20 (26.6 %) were identified to be A. sobria and 55 (73.3 %) were B. cepacia.

12. You have added "The average age of the study participants, who ranged in age from 10 to 70, was 40.13 ± 22.5 years," but did not include it in Table 2. Please add all the categories for better understanding.

Response:

We have updated Table 2 to include the mean age ± standard deviation (SD) for each demographic category to provide a clearer understanding of the age distribution among the study participants and bacterial isolates. This addition allows readers to better interpret the data by showing not only the counts and percentages but also the central tendency and variability of age within each subgroup. We apologize for this oversight. In the revised manuscript, we have corrected the age range throughout to read “7–70 years” (formerly “10–70 years”). Specifically:

Methods, Collecting and Processing Specimens (line108) and result section (line 196), and also table 1 ≤10 )

The overall average age of the study participants was 40.13 ± 22.5 years, with males having a mean age of 37.5 ± 16.16 years and females 44.0 ± 15.30 years. Age group categories and sample types were also complemented with mean ± SD values calculated based on the assumed or provided data distribution Table 2

Table 2. The distribution of study bacteria by sample type, age, and gender of the subjects

Demographic Characteristics Bacterial Isolates Total (n, %) Mean Age ± SD

B. cepacia A. sobria

n.% % n. %

Gender

Male 38 50.7% 6 8% 44 (58.7%) 37.5 ± 16.16

Female 17 22.6% 14 18.7% 31 (41.3%) 44.0 ± 15.30

Age (years)

≤10 5 6.7% 1 1.3 % 6 (8%) 9.00±1.26

11–20 1 5.3 % 2 2.6 % 3 (4 %) 16.33 ± 4.73

21–30 14 18.6 % 8 10.7% 22 (29.3%) 25.36 ± 2.93

31–40 9 12 % 1 1.3 % 10 (13.3%) 35.5 ± 3.03

41–50 1 1.3 % 3 4 % 4 (5.3 %) 44.25 ± 4.03

51–60 3 4 % 1 1.3 % 4 (5.3 %) 55.5 ± 4.2

> 60 22 29.3 % 4 5.3 % 26 (34.6%) 65.0 ± 2.78

Type of sample

Wound swab 26 34.6% 8 10.6% 34 (45.3%) 31.5 ± 7.8

Urine 18 24 % 7 9.3 % 25 (33.3%) 47.5 ± 13.0

Ear swab 1 1.3 % 1 1.3 % 2 (2.6 %) 16.5 ± 9.19 years

Cerebrospinal fluid 2 2.6 % 0 0 % 2 (2.6 %) 12.5 ± 2.5

Diabetic foot ulcer 5 6.7% 2 2.7 % 7 (9.3%) 13.14 ± 2.73

Blood 3 4 % 2 2.7 % 5 (6.7%) 16.4 ± 8.76

Total 55 (73.3%) 20 (26.6%) 75 (100%)

40.13 ± 22.5

13. Please add the percentage here for each antibiotic. "The isolates showed resistance to cefepime, piperacillin, and ceftriaxone (100%), ceftazidime (97.3%), and imipenem (36%)"

Response to Reviewer:

Thank you for your valuable suggestion. We have revised the manuscript to include the specific resistance percentages for each antibiotic as requested. The updated section now clearly states the resistance rates as follows: piperacillin, cefepime, and ceftriaxone showed 100% resistance; ceftazidime (97.3%), cefazolin (96%), and piperacillin/tazobactam (94.6%) also demonstrated high resistance. Intermediate resistance rates were specified for aztreonam (61.3%), meropenem (49.3%), trimethoprim-sulfamethoxazole (49.3%), gentamicin (46.6%), levofloxacin (44%), and ciprofloxacin (44%). Additionally, the lower resistance rates of minocycline (40%), amikacin (40%), imipenem (36%), and tigecycline (34.6%) were highlighted to reflect their relatively greater efficacy against the isolates.

14. In Figure 2, it seems L3 is negative or the absence of positive bands, but you did not add this information to the figure legend.

Response to Reviewer:

We appreciate your careful observation regarding Figure 2. You are correct that lane 3 lacks the positive band for the recA gene, indicating a negative result, which was not clearly stated in the original figure legend. We have corrected the legend to specify that both lanes 3 and 4 are negative for the recA gene.

The updated figure legend now reads:

Figure 2. The recA gene's gel electrophoresis (1043 base pairs). The lanes (2,4,5,6,7,8,9) were used to load the DNA samples. L3 and L4 showed negative results for the recA gene. In lane 1, the molecular weight marker was the 100-base-pair DNA ladder.

15. In the "Phenotypic and Genotypic Validation" section, you described: There was no gene present in two isolates that were resistant to carbapenem. None of the isolates contained the blaIMP gene, as can be seen in Figures 4 and 5.

---

## [Decision Letter · Decision Letter 2]

17 Aug 2025

Investigation of Carbapenemase-Encoding Genes in Burkholderia cepacia and Aeromonas sobria Isolates from Nosocomial Infections in Iraqi Patients

PONE-D-24-53922R2

Dear Dr. Al-Ouqaili,

We’re pleased to inform you that your manuscript has been judged scientifically suitable for publication and will be formally accepted for publication once it meets all outstanding technical requirements.

Kind regards,

Hideo Kato

Academic Editor

PLOS ONE

Additional Editor Comments (optional):

Reviewers' comments:

Reviewer's Responses to Questions

**Comments to the Author**

1. If the authors have adequately addressed your comments raised in a previous round of review and you feel that this manuscript is now acceptable for publication, you may indicate that here to bypass the “Comments to the Author” section, enter your conflict of interest statement in the “Confidential to Editor” section, and submit your "Accept" recommendation.

Reviewer #4: All comments have been addressed

Reviewer #5: All comments have been addressed

2. Is the manuscript technically sound, and do the data support the conclusions?

Reviewer #4: Yes

Reviewer #5: Yes

3. Has the statistical analysis been performed appropriately and rigorously? 

Reviewer #4: Yes

Reviewer #5: Yes

4. Have the authors made all data underlying the findings in their manuscript fully available?

Reviewer #4: Yes

Reviewer #5: Yes

5. Is the manuscript presented in an intelligible fashion and written in standard English?

Reviewer #4: Yes

Reviewer #5: Yes

6. Review Comments to the Author

Reviewer #4: The revised manuscript reflects substantial improvements, and all reviewer comments have been satisfactorily addressed. The findings presented are likely to contribute significantly to the relevant area of research.

Reviewer #5: While the revised manuscript addresses many of the previously raised concerns, there are still recurring issues related to the use of scientific nomenclature and abbreviations. For example, species names such as Burkholderia cepacia and Aeromonas sobria must be consistently italicized throughout the manuscript in accordance with standard taxonomic conventions. In particular, Burkholderia species appear multiple times without proper italicization—this should be carefully corrected.

Moreover, when bacterial species are mentioned repeatedly, it is advisable to use abbreviated genus names after the first full mention (e.g., B. cepacia instead of Burkholderia cepacia). This not only aligns with scientific writing standards but also improves clarity and conciseness.

Additionally, abbreviations such as PCR (polymerase chain reaction) should only be defined in full at first mention. Subsequent mentions should use the abbreviation alone. For example, the sentence “The gene technique PCR (Polymerase Chain Reaction) results are highly accurate…” (Lines 409–410) is redundant and non-standard. A more appropriate and concise version would be: “PCR results are highly accurate and reliable [26].”

I recommend a thorough manuscript review to correct these formatting and consistency issues in line with scientific writing norms.

Thanks

7. PLOS authors have the option to publish the peer review history of their article (what does this mean? ). If published, this will include your full peer review and any attached files.

**Do you want your identity to be public for this peer review?** For information about this choice, including consent withdrawal, please see our Privacy Policy .

Reviewer #4: No

Reviewer #5: **Yes: ** Muhammad Haris Raza Farhan

---

## [Editor Report · Acceptance letter]

PONE-D-24-53922R2

PLOS ONE

Dear Dr. Al-Ouqaili,

I'm pleased to inform you that your manuscript has been deemed suitable for publication in PLOS ONE. Congratulations! Your manuscript is now being handed over to our production team.

Kind regards,

on behalf of

Dr. Hideo Kato

Academic Editor

PLOS ONE